# Induced pluripotent stem cells of endangered avian species

Masafumi Katayama [1✉], Tomokazu Fukuda [2✉], Takehito Kaneko [2], Yuki Nakagawa[2], Atsushi Tajima[3], Mitsuru Naito[4], Hitomi Ohmaki[5], Daiji Endo[5], Makoto Asano[6], Takashi Nagamine[7], Yumiko Nakaya[7], Keisuke Saito[8], Yukiko Watanabe[8], Tetsuya Tani[9], Miho Inoue-Murayama[10], Nobuyoshi Nakajima[1] & Manabu Onuma [1,5✉]

The number of endangered avian-related species increase in Japan recently. The application of new technologies, such as induced pluripotent stem cells (iPSCs), may contribute to the recovery of the decreasing numbers of endangered animals and conservation of genetic resources. We established novel iPSCs from three endangered avian species (Okinawa rail, Japanese ptarmigan, and Blakiston's fish owl) with seven reprogramming factors (*M3O*, *Sox2*, *Klf4*, *c-Myc*, *Nanog*, *Lin28*, and *Klf2*). The iPSCs are pluripotency markers and express pluripotency-related genes and differentiated into three germ layers in vivo and in vitro. These three endangered avian iPSCs displayed different cellular characteristics even though the same reprogramming factors use. Japanese ptarmigan-derived iPSCs have different biological characteristics from those observed in other avian-derived iPSCs. Japanese ptarmigan iPSCs contributed to chimeras part in chicken embryos. To the best of our knowledge, our findings provide the first evidence of the potential value of iPSCs as a resource for endangered avian species conservation.

[1] Biodiversity Division, National Institute for Environmental Studies, 16-2 Onogawa, Tsukuba, Ibaraki 305-8506, Japan. [2] Graduate School of Science and Engineering, Iwate University, 4-3-5 Ueda, Morioka, Iwate 020-8551, Japan. [3] Faculty of Life and Environmental Sciences/T-PIRC, University of Tsukuba, 1-1-1 Ten-noh Dai, Tsukuba, Ibaraki 305-8572, Japan. [4] National Institute of Agrobiological Sciences, 2-1-2 Kannondai, Tsukuba, Ibaraki 305-8602, Japan. [5] School of Veterinary Medicine, Rakuno Gakuen University, 582 Bunkyodai Midorimachi, Ebetsu, Hokkaido 069-8501, Japan. [6] Faculty of Applied Biological Sciences, Gifu University, 1-1 Yanagido, Gifu 501-1193, Japan. [7] Okinawa Wildlife Federation, 308-7-205 Maehara, Uruma, Okinawa 904-2235, Japan. [8] Institute for Raptor Biomedicine Japan (Kushiro Shitsugen Wildlife Center), 2-2101 Hokuto, Kushiro, Hokkaido 084-0922, Japan. [9] Department of Agriculture, Kindai University, 3327-204 Nakamachi, Nara 631-0052, Japan. [10] Wildlife Research Center, Kyoto University, 2-24 Tanaka-Sekiden-Cho, Sakyo-Ku, Kyoto 606-8203, Japan. ✉email: katayama.masafumi@nies.go.jp; tomofukuda009@gmail.com; monuma@nies.go.jp

The land of Japan extends from north to south, resulting in a huge diversity of climates and environmental conditions. Due to their adaptation to diverse climates, various endemic species have evolved in Japan. Japan is thus a hotspot of animal diversity. The number of endangered species has been increasing in recent years, mainly due to human activities such as forest destruction and global warming. According to data from the Ministry of the Environment Government of Japan, 1446 species are categorized as endangered (red list category CR, EN, and VU) (http://www.env.go.jp/press/113666.pdf). Notably, 98 avian species are categorized as endangered avian-related species, including 24 critically endangered, 31 endangered, and 43 vulnerable species. The conservation of endangered species for the next generation is essential to maintain biological diversity for succeeding generations.

Cultured cells have the potential as a bioresource for the conservation of endangered animals. Recently, new technologies have been developed that control the differentiation status of cells, such as induced pluripotent stem cells (iPSCs). The technology of iPSCs enables the status of cells to be changed from somatic cells to PSCs. The latter can differentiate into terminally differentiated cells.

In this study, we focused on three endangered avian species endemic to Japan: Okinawa rail (*Hypotaenidia okinawae*), Japanese ptarmigan (*Lagopus muta japonica*), and Blakiston's fish owl (*Bubo blakistoni*). To the best of our knowledge, the establishment of iPSCs derived from endangered avian species has not been reported. Although a chicken embryonic germ cell line was established in a previous study on chicken primordial germ cells, optimized conditions for establishing avian iPSCs have not been fully characterized, even in domestic avian species, such as chicks and quail[1–3]. Recently, Kim et al. reported that iPSC-like cells could generate adult chicken feather follicle cells, which are one of the most easily accessible cell sources for avians, suggesting that iPSC-like cells can be established from various types of somatic cells[4]. Our previous study efficiently established chick-derived iPSCs using a modified *Oct3/4* transcriptional factor, which involves the transcriptional activation domain of *MyoD* along with other reprogramming genes (*Sox2*, *Klf4*, *c-Myc*, *Nanog*, and *Lin28*)[5]. However, it has not been clear whether our improved reprogramming method is sufficient to reprogram the somatic cells of other avian species, including endangered Japanese species.

In this study, to the best of our knowledge, we succeeded in establishing for the first time iPSCs from primary fibroblasts of the three aforementioned endangered avian species. Furthermore, we tried to address the biological nature of the established cells, even though genome information is poorly addressed in these species. Our study provides useful information for understanding avian stem cell biology.

## Results

### Obtaining primary cells from endangered avians.
We established primary fibroblasts derived from dead Okinawa rail and dead Japanese ptarmigan, and emerging pinfeather of Blakiston's fish owl (Fig. 1a, b). Emerging pinfeather can fall off birds with the application of small mechanical pressure when the birds are held during labeling with identification tags. We attempted to obtain somatic cells from emerging pinfeathers. As an example, a collection of emerging pinfeathers from a Japanese golden eagle is displayed in Fig. 1bi. Feather regrowth in Japanese golden eagles usually occurs in late spring (Fig. 1bii). We used the jelly-like tissue inside the emerging pinfeather as the starting material for primary cell culture (Fig. 1biv, v). Somatic cells were efficiently obtained from pinfeathers with a 100% high success rate (six of

six samples). Using this method, we obtained somatic cells from other avian species (Fig. 1c). To analyze the biological characteristics of the pinfeather-derived cells, we compared the distribution patterns of F-actin and vimentin, and the cell cycle of Japanese golden eagle cells with those of chicken embryonic fibroblasts (Fig. 1d, e). No critical difference was observed between pinfeather-derived cells and chick embryonic fibroblasts. Therefore, we used the same culture methods for dead avian-derived cells and emerging pinfeather-derived cells in this study.

### Generation of iPSCs derived from endangered avian species, chicken, and mouse with a PB-TAD-7F reprogramming vector.
Fibroblasts from Okinawa rail, Japanese ptarmigan, and Blakiston's fish owl were cryopreserved in a liquid nitrogen tank for approximately 8 to 12 years (Fig. 1f and Supplementary Fig. 1). As previously explained, we attempted to establish iPSCs from three endangered avian species (Okinawa rail, Japanese ptarmigan, and Blakiston's fish owl), as well as mouse- and chicken-derived iPSCs. First, we attempted to establish avian iPSCs with PB-R6F (modified *Oct3/4* ($M_3O$), *Sox2*, *Klf4*, *c-Myc*, *Lin28*, and *Nanog*) (Supplementary Methods). This system can be used to establish chicken iPSCs[5]. Although Okinawa rail, Japanese ptarmigan, chicken, and mouse primary colonies of iPSCs allowed us to generate the PB-R6F reprogramming vector. However, Blakiston's fish owl-derived primary colonies did not appear (Supplementary Methods, Supplementary Figs. 2–4). Therefore, to establish these species-derived iPSCs, we improved the poly cistronic single all-in-one vector (PB-TAD-7F) encoding modified *Oct3/4* ($M_3O$), *Sox2*, *Klf4*, *c-Myc*, *Klf2*, *Lin28*, and *Nanog*, as shown in Fig. 1g. The process to establish iPSCs is shown in Fig. 1h. We established iPSCs using the PB-TAD-7F reprogramming vector once for Blakiston's fish owl, twice for Okinawa rai, and three times for Japanese ptarmigan (Supplementary Fig. 2). We introduced the expression plasmid into primary cells using the lipofection method. However, the success rate was only approximately 5% or lower. Therefore, we added hygromycin for the selection of reprogramming gene-integrated cells. After re-seeding the transfected cells onto the feeder plate, we observed iPS-like primary colonies in mice, chicken, Okinawa rail, Japanese ptarmigan, and Blakiston's fish owl-derived cells (Fig. 1i). In avians, at least 20 primary colonies of chicken, Okinawa rail, and Japanese ptarmigan were observed in six-well plates. The number of primary colonies of Blakiston's fish owl was only seven. After picking colonies, we established iPSC clones derived from fibroblasts of mouse, chicken, Okinawa rail, Japanese ptarmigan, and Blakiston's fish owl (Fig. 1j). The establishment efficiency of iPSCs with PB-TAD-7F was approximately 42–100% in avian cells (Supplementary Fig. 2). The iPSC colony morphology of Japanese ptarmigans was three-dimensional, whereas Blakiston's fish owl iPSC colonies displayed a flat morphology. Okinawa rail iPSC colonies exhibited an intermediate morphology between those of Japanese ptarmigan and Blakiston's fish owl (Fig. 1j). These iPSCs were stably passaged more than 20 times.

### Detection of reprogramming vector genes, staining with alkaline phosphatase (AP) and stage-specific embryonic antigen (SSEA), expression of Tert, and cell growth analysis.
The expected specific PCR products were amplified only in iPSCs (Fig. 2a and Supplementary Fig. 5), but not in parent fibroblasts. We concluded that the PB-TAD-7F reprogramming vector was successfully introduced into the genome of iPSCs.

All our established iPSC lines exhibited high AP expression (Fig. 2b). Furthermore, at least one marker of SSEA-1 or SSEA-3 was positive in each of our established iPSC lines (Fig. 2c, d).

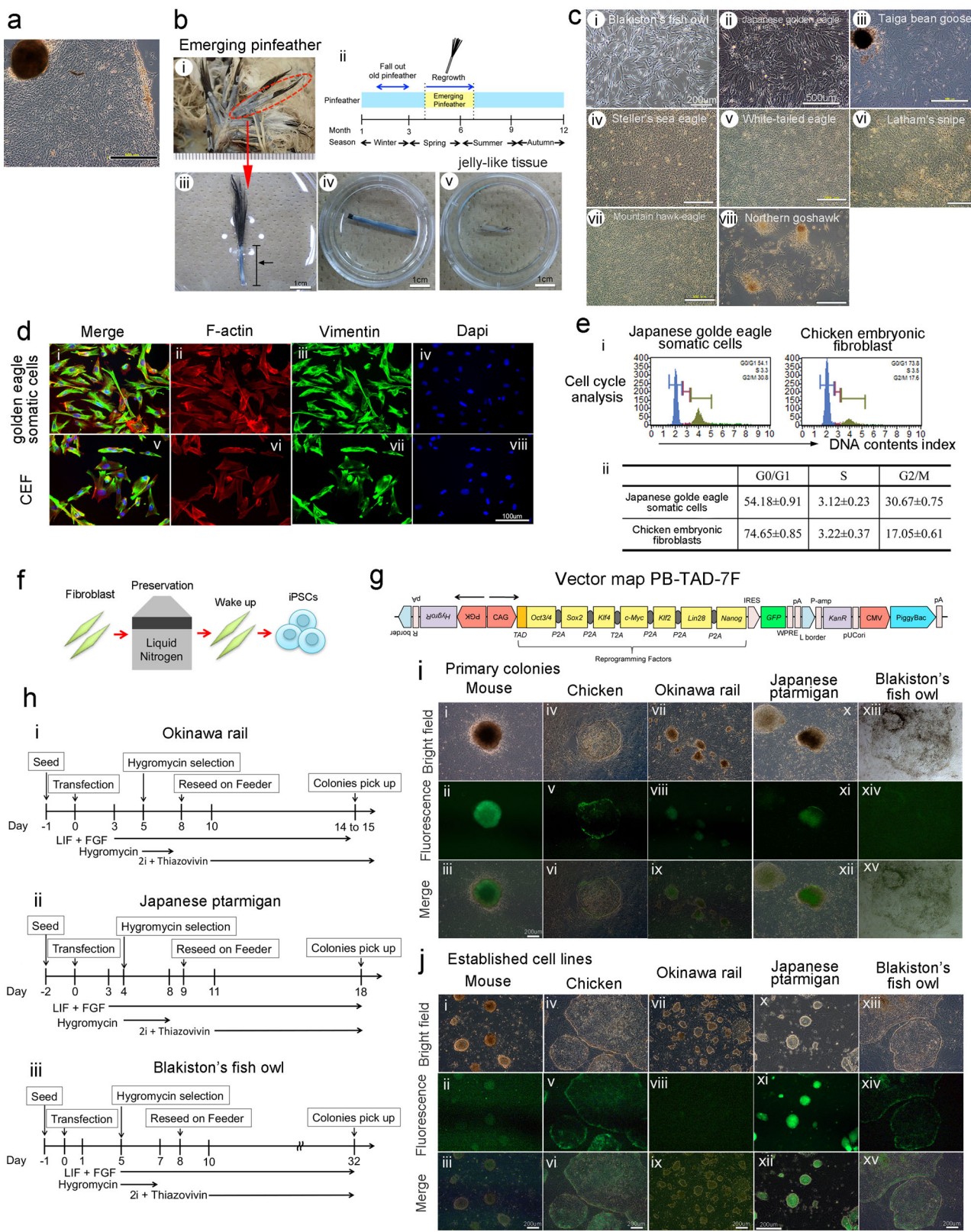

SSEA-4 was never detected (Supplementary Fig. 6), indicating that our established iPSC lines express multiple pluripotent markers.

Telomere elongation is activated after successful iPSC transformation[6]. *Tert* gene expression in iPSCs was significantly higher than the expression in fibroblasts using real-time PCR (Fig. 2e). The expression level of *Tert* in mouse iPSCs was almost identical to that in parent fibroblasts (Fig. 2e). This observation agreed with prior results[7,8].

We then sequential passaged the established Okinawa rail, Japanese ptarmigan, and mouse iPSC lines in single-cell conditions after digestion with Accutase (Fig. 2f). All three iPSC types continued to proliferate until a population doubling (PD) value of approximately 70 (Fig. 2g). Our Blakiston's fish owl and

**Fig. 1 Collection of somatic cells from Japanese endangered avian species and establishment of iPSCs derived from Japanese endangered avians, chicken, and mouse. a** Cell morphology of Okinawa rail-derived primary fibroblasts from the muscle tissue. The bars represent 500 μm. **b** Collection of somatic cells from emerging pinfeathers, and the cycle of pinfeathers of Japanese golden eagle and sampling point. Japanese golden eagle-derived pinfeathers (i). The cycle of emerging pinfeathers of Japanese golden eagle from late spring to early summer during one year, and sampling point (ii). Japanese golden eagle-derived pinfeathers (iii). The arrow indicates the feather sheath of the emerging pinfeather. Emerging pinfeathers after trimming for cell culture (iv). Jelly-like tissues were obtained from the inside of the emerging pinfeather (v). The bars represent 1 cm. **c** Somatic cells from the pinfeathers of wild avian species. Blakiston's fish owl (i), Japanese golden eagle (ii), Taiga bean goose (iii), Steller"s sea eagle (iv), White-tail eagle (v), Latham's snipe (vi), Mountain hawk-eagle (vii), and Northern goshawk (vii) derived somatic cells. The names of the avian species are presented in each image. The bars represent 200 μm (Blakiston's fish owl) and 500 μm (other avian species). **d** F-actin and vimentin staining of Japanese golden eagle and chicken primary cells. Primary cells from the Japanese golden eagle pinfeather (i–iv) and chick embryo fibroblasts are displayed (v–viii). Merged (i and v), F-actin-stained (ii and vi), vimentin-stained (iii and vii), and DAPI stained (iv and viii) images are shown. Bars indicate 100 μm. **e** Cell cycle analysis of primary cells from Japanese golden eagles and chicken embryos. Somatic cells from the emerging pinfeather of a Japanese golden eagle and chick embryonic fibroblasts (i). Cell cycle ratio of Japanese golden eagle somatic cells and chicken embryonic fibroblasts (ii). The reported data of the table were the mean values ± S.D. **f** Experimental procedure used to establish Okinawa rail, Japanese ptarmigan, and Blakiston fish owl-derived iPSCs. Somatic cells derived from the endangered birds were collected, and preserved in liquid nitrogen tanks. **g** Vector map of the PB-TAD-7F reprogramming vector. **h** Establishment of Okinawa rail (i), Japanese ptarmigan (ii), and Blakiston fish owl (iii) iPSCs. We established iPSCs from Okinawa rail, Japanese ptarmigans and Blakiston fish owl somatic cells according to this flow. **i** Morphologies of primary colonies of iPSCs derived from mouse, chicken, Okinawa rail, Japanese ptarmigan, and Blakiston's fish owls. Bright field images (i, iv, vii, x, and xiii), green fluorescence protein (GFP) images (ii, v, viii, xi, and xiv), and merged images (iii, vi, ix, xii, and xv) represent in this image. Mouse images (i–iii), chicken images (iv–vi), Okinawa rail images (vii–ix), Japanese ptarmigan images (x–xii), and Blakiston's fish owl images (xiii–xv). The bars represent 200 μm. **j** Cell morphology of established iPSC lines derived from mouse, chicken, Okinawa rail, Japanese ptarmigan, and Blakiston's fish owl. Bright field images (i, iv, vii, x, and xiii), green fluorescence protein (GFP) images (ii, v, viii, xi, and xiv), and merged images (iii, vi, ix, xii, and xv) represent in this image. Mouse images (i–iii), chicken images (iv–vi), Okinawa rail images (vii–ix), Japanese ptarmigan images (x–xii), and Blakiston's fish owl images (xiii–xv). The bars represent 200 μm. Those images shows the mouse iPSCs of day 6 at passage 2 show (i–iii), the chicken iPSCs of day 5 at passage 2 (iv–vi), the Okinawa rail iPSCs of day 3 at passage 3 (vii–ix), the Japanese ptarmigan iPSCs of day 4 at passage 9 (x–xii), and the Blakiston's fish owl iPSCs of day 7 at passage 10 (xiii–xv).

chicken iPSC lines caused extensive cell death after enzymatic digestion. In contrast, sequential mechanical digestion passages without enzymatic treatment could be performed (Fig. 2f). The mechanical method allowed us to reach at least passage 20 for both Blakiston's fish owl and chicken iPSC lines (Supplementary Fig. 7). These data indicate that Okinawa rail, Japanese ptarmigan, Blakiston's fish owl, chicken, and mouse iPSC lines stably maintained the characteristics of stem cells.

**DNA contents and karyotype analyses of three endangered avian iPSCs.** Next, we analyzed the chromosomes of Okinawa rail, Japanese ptarmigan, and Blakiston fish owl iPSCs. To evaluate chromosome stability, flow cytometry was first performed. As shown in Supplementary Figs. 8a, b, Okinawa rail, Japanese ptarmigan, and Blakiston's fish owl-derived diploid chromosome (2n) iPSCs were evident. Karyotype analysis revealed that the chromosome numbers of Okinawa rail, Japanese ptarmigan, and Blakiston's fish owl iPSCs were approximately 70–84, 80–89, and 76–82, respectively Supplementary Fig. 8c, Supplementary Table 1). Thus, we confirmed that the three derived iPSCs could maintain diploid conditions (2n).

**mRNA expression levels of pluripotency-related genes in mouse, chicken, Okinawa rail, Japanese ptarmigan, and Blakiston's fish owl iPSCs.** *Oct3/4*, *Sox2*, and *Nanog* are members of the core circuit of pluripotency-associated transcription factors in mouse pluripotent stem cells[9]. *POU5* (mammalian: *Oct3/4*) and *Nanog* were highly expressed in Okinawa rail, Japanese ptarmigan, Blakiston's fish owl, and chicken iPSCs compared to the expression in parent fibroblasts (Fig. 3a). Although *Sox2* is also a core pluripotency gene in mouse PSCs, species differences existed among Okinawa rail, Japanese ptarmigan, Blakiston's fish owl, and chicken iPSCs concerning *Sox family transcription factor 2* (*Sox2*) expression levels (Fig. 3a). In contrast to *Sox2*, the expression level of *Sox3* was commonly increased in these avian iPSCs (Fig. 3a). Therefore, the *sox* family of genes also plays an important role in avian iPSCs. *Lin28* is also a major transcription factor that maintains stem cell pluripotency[10]. The expression

levels of *Lin28a* and *Lin28b* increased after conversion to iPSCs in the Okinawa rail, Japanese ptarmigan, Blakiston's fish owl, and chicken iPSCs (Fig. 3a). These data indicate that the four established iPSCs strongly expressed *POU5*, *Nanog*, *Lin28*, and *Sox family* transcription factors. All these factors are directly related to pluripotency.

To evaluate candidate genes involved in maintaining pluripotency of avian stem cells, we analyzed the expression of *CDH1*, *Fbxo15*, *Esrrb*, *Sall4*, *Dnm3tb*, and *Rex1*. The activities of these genes were increased in our established mouse, chicken, Okinawa rail, Japanese ptarmigan, and Blakiston's fish owl iPSCs when compared to the activities in fibroblasts (Fig. 3b). In addition to these transcription factors, the *Nanog-like* gene (which is an avian pluripotency marker, whereas does not exist in mouse) was also strongly expressed in our avian iPSCs. The findings indicate that the established Okinawa rail, Japanese ptarmigan, Blakiston's fish owl, and chicken iPSCs highly express the major genes of maintaining the pluripotency of avian stem cells.

The expression levels of other reprogramming-associated genes, including *Klf4*, *Klf2*, *Tfcp2l1*, *Tbx3*, *c-Myc*, and *Prdm14*, differed from the expression pattern of mouse iPSCs (Fig. 3c). Thus, the gene expression patterns of avian iPSCs did not fully agree with those of mouse iPSCs. Furthermore, compared to fibroblasts, chicken iPSCs showed elevated expression of germ cell marker genes, such as *Vasa* and *Dazl* (Fig. 3d). Similar to chicken iPSCs, Okinawa rail, Japanese ptarmigan, and Blakiston's fish owl iPSCs showed increased gene expression levels of *Vasa* (Fig. 3d). Elevation of the *Dazl* gene was not observed in the Japanese ptarmigan (Fig. 3d). These data are evidence of species differences in germ cell marker expression among avian iPSCs.

**mRNA expression of leukemia inhibitory factor (Lif), Lif receptor (Lifr), fibroblast growth factor (Fgf) family, and Wnt family, and cell growth dependency of LIF, FGF, 2i, and serum in Okinawa rail, Japanese ptarmigan, and Blakiston's fish owl iPSCs.** The expression patterns of *Lif*, *Lifr*, *Fgf3*, *Fgf8*, and *Fgf10* in chicken, Okinawa rail, Japanese ptarmigan, and Blakiston fish owl iPSCs were consistent with those in the mouse iPSCs

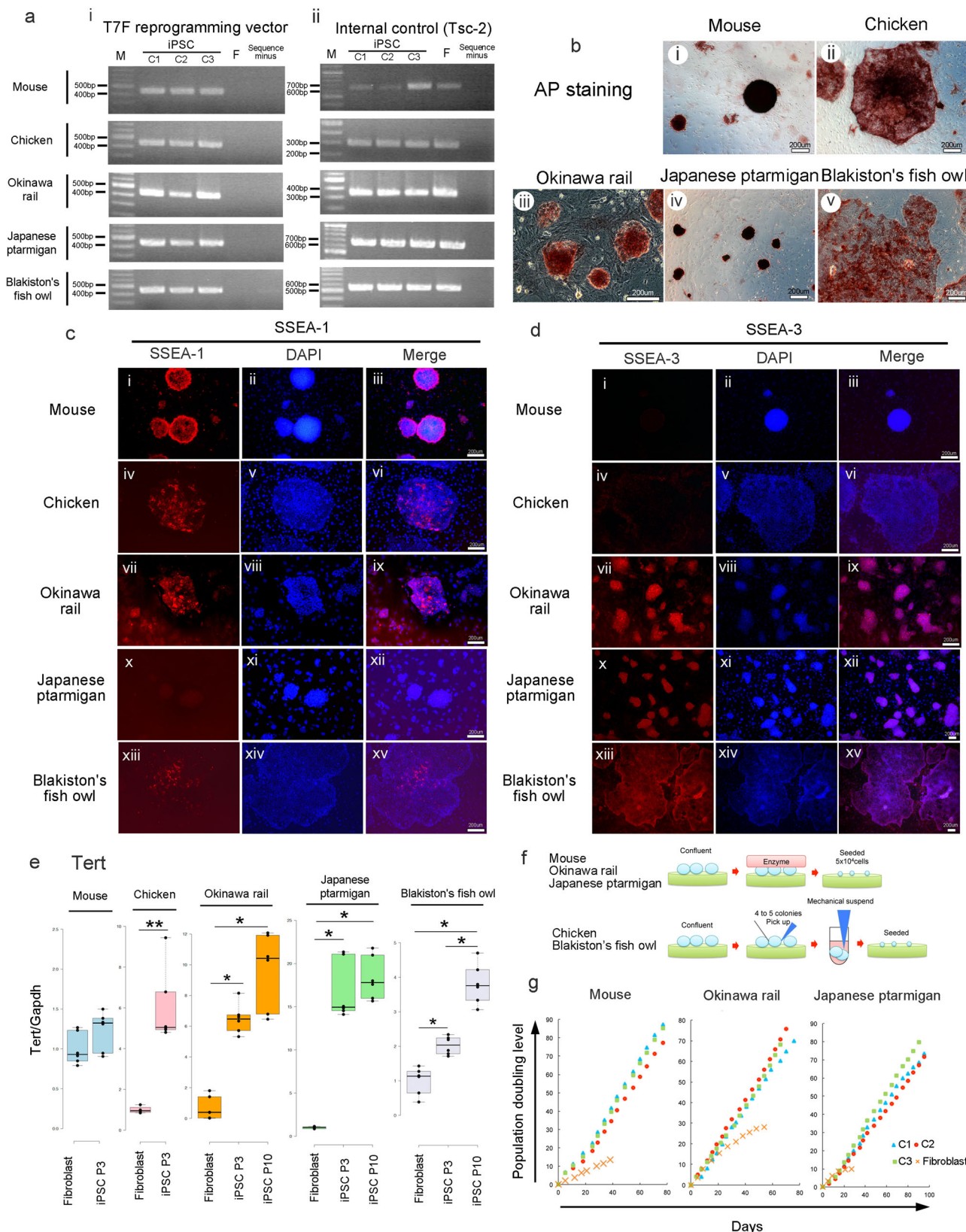

(Supplementary Fig. 9a, b). In contrast to *Lif* and *Fgf*, the expression of the *Wnt3a* canonical pathway signal and *Wnt5a* non-canonical pathway signal differed between avian iPSCs and mouse iPSCs (Supplementary Fig. 9c).

Next, we evaluated whether the addition of the LIF and FGF growth factors was essential for the maintenance of Okinawa rail,

Japanese ptarmigan, and Blakiston's fish owl-derived iPSCs. In all three of the derived iPSCs, the cell number and AP staining activity did not differ between cells cultured in normal medium and medium lacking LIF (Supplementary Fig. 9d). In medium lacking FGF, although Okinawa rail and Japanese ptarmigan-derived iPSCs maintained their pluripotency, similar to that

**Fig. 2 Characteristics of the five iPSCs. a** Detection of reprogramming vectors using genomic PCR in mouse, chicken, Okinawa ral, Japanese ptarmigan, and Blakiston's fish owl iPSCs. The PB-TAD-7F vector (i) and the Tsc2 internal control (ii) were detected. Lane M, molecular weight marker; lane C1, iPSC clone1, lane C2, iPSC clone 2; lane C3, iPSC clone 3; lane F, fibroblast; and lane sequence minus, PCR with sequence minus. **b** Detection of alkaline phosphatase (AP) activity in mouse (i), chicken (ii), Okinawa rail (iii), Japanese ptarmigan (iv) and Blakiston's fish owl (v) iPSCs. The bars represent 200 μm. Detection of stage-specific embryonic antigen (SSEA)-1 (**c**) and -3 (**d**) in mouse, chicken, Okinawa rail, Japanese ptarmigan, and Blakiston's fish owl iPSCs. SSEA-1 (i, iv, vii, x, and xiii), DAPI (ii, v, vii, xi, and xiv) and merged (iii, vi, ix, xii, and xv) image represent in (**c**), SSEA-3 (i, iv, vii, x, and xiii), DAPI (ii, v, vii, xi, and xiv) and merged (iii, vi, ix, xii, and xv) image represent in (**d**). The bars represent 200 μm. **e** Detection of telomerase reverse transcriptase (Tert) expression by real-time PCR. Tert expression was quantified relative to the *GAPDH* internal control. The fibroblast expression level was 1.0. Light blue bars are mouse; pink bars are chicken; orange bars are Okinawa rail; light green bars are Japanese ptarmigan; and light purple bars are Blakiston's fish owl-derived cells. Mouse and chicken: The bars represent fibroblasts and iPSCs (passage number 3; P3). Okinawa rail, Japanese ptarmigan, and Blakiston's fish owls: bars represents fibroblasts, iPSCs (P3) and iPSCs (P10). Centerlines of box plots indicate medians; box limits indicate the 25th and 75th percentiles as determined by BoxPlotR (http://shiny.chemgrid.org/boxplotr/). $n = 4$ (chicken fibroblast), $n = 6$ (other). *$P < 0.05$, **$P < 0.01$. **f** Passaging of mouse, chicken, Okinawa rail, Japanese ptarmigan, and Blakiston's fish owl-derived iPSCs. **g** Cell growth analysis of established mice, Okinawa rail, and Japanese ptarmigan-derived iPSCs. The blue triangles represent iPSCs clone 1, red circles represent iPSCs clone 2, green squares represent iPSCs clone 3, and orange circles represent fibroblasts. We seeded $5 \times 10^4$ iPSCs or fibroblasts on the culture plate and harvested them when growth was confluent.

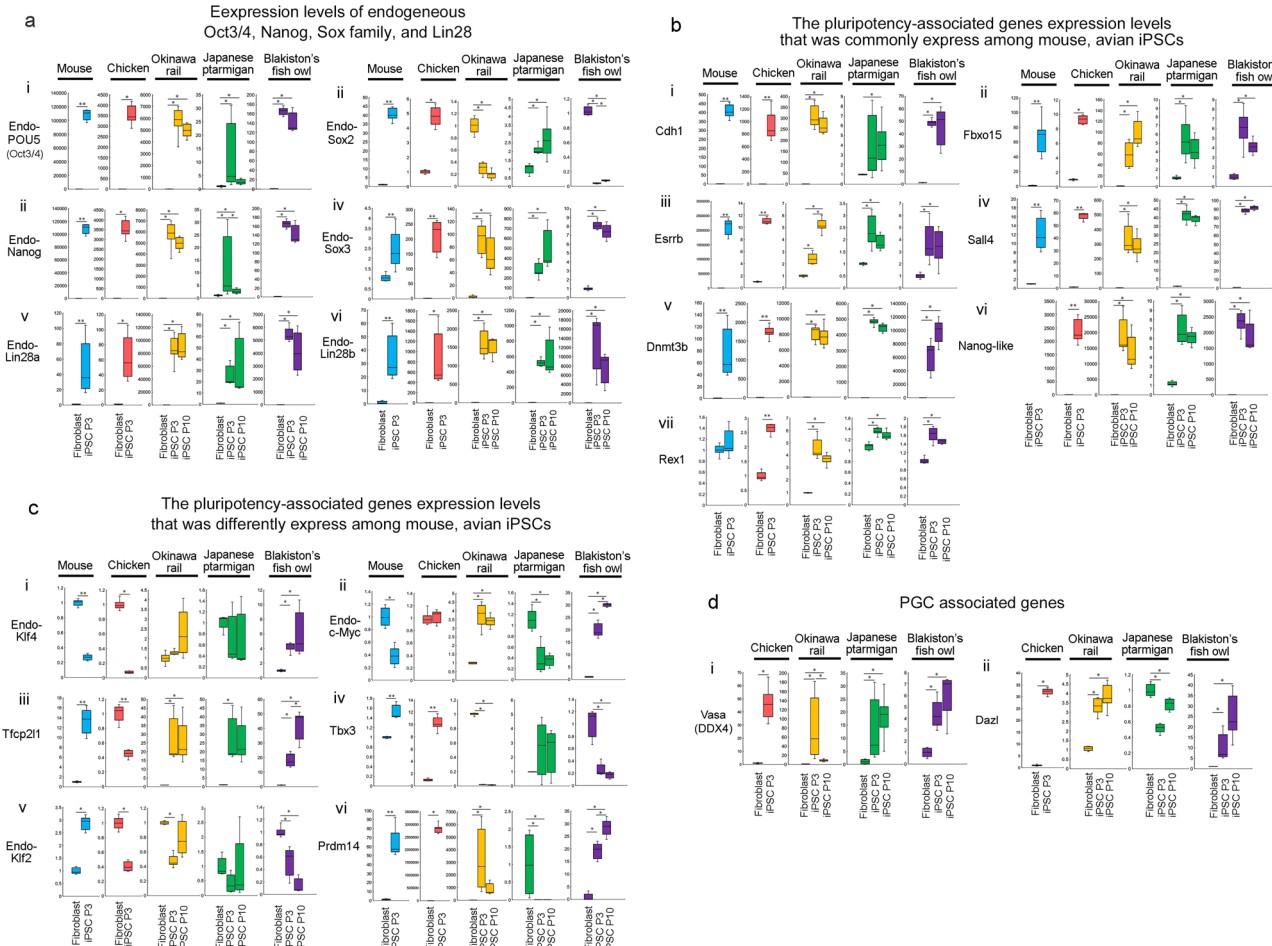

**Fig. 3 Analysis of pluripotency-related gene expression in mouse, chick, Okinawa rail, Japanese ptarmigan, and Blakiston's fish owl-derived iPSCs and parent fibroblasts. a** mRNA expression levels of *POU5* (mammalian *Oct3/4*), Nanog, Sox family, and *Lin28* in mouse, chicken, Okinawa rail, Japanese ptarmigan, and Blakiston's fish owl-derived iPSCs and fibroblasts. endo-*POU5* (mammalian *Oct3/4*) (i), endo-*Sox2* (ii), endo-*Nanog* (iii), *Sox3* (iv), endo-*Lin28a* (v), and endo-*Lin28b* (vi) represents. **b** Pluripotency-associated genes commonly expressed in mouse, chick Okinawa rail, Japanese ptarmigan, and Blakiston fish owl iPSCs. *Cdh1* (i), *Fbxo15* (ii), *Esrrb* (iii), *Sall4* (iv), *Dnmt3b* (v), *Nanog-like* (vi), and *Rex1* (vii) represents. **c** Pluripotency-associated genes differentially expressed in mouse, chick Okinawa rail, Japanese ptarmigan, and Blakiston fish owl iPSCs. Endo-*Klf4* (i), Endo-*c-Myc* (ii), *Tfcp2l1* (iii), *Tbx3* (iv), Endo-*Klf2* (v), and *Prdm14* (vi) represents. **d** mRNA expression levels of primordial germ cell (PGC) marker genes. *Vasa* (i), *Dazl* (ii) represents. Gene expression was quantified relative to the *GAPDH* internal control. The fibroblast expression level was 1.0. Blue bars, mouse; red bars, chicken; yellow bars, Okinawa rail; green bars, ptarmigan; and purple bars, Blakiston's fish owl-derived cells. Mouse and chicken: Fibroblasts and the iPSCs (passage number 3; P3) represents. Okinawa rail, Japanese ptarmigan, and Blakiston's fish owl: Fibroblasts, iPSCs (P3), and iPSCs (P10) represents. Centerlines of box plots indicate medians; box limits indicate the 25th and 75th percentiles. $n = 4$ (Chicken fibroblast in *Pou5*, *Sox2*, *Nanog*, *Lin28a*, *Lin28b*, *Cdh1*, *Fbxo15*, *Esrrb*, *Klf4*, *Tbx3*, *Klf2*, *Prdm14*, *Vasa*, and *Dazl*. Okinawa rail fibroblast in *Sox2*, *Esrrb*, *Klf4*, *Tbx3*, and *Klf2*.), $n = 6$ (other). *$P < 0.05$, **$P < 0.01$.

observed in the control medium, Blakiston's fish owl-derived iPSCs partially differentiated (Supplementary Fig. 9d). In medium without two inhibitors (2i), Okinawa rail and Japanese ptarmigan iPSCs maintained AP activity, whereas Blakiston fish owl iPSCs had decreased AP activity (Supplementary Fig. 9d). We then evaluated whether serum was essential for the culture of Okinawa rail, Japanese ptarmigan, and Blakiston fish owl-derived iPSCs. Serum contains various growth factors, such as insulin-like growth factor 1 (IGF1) and epidermal growth factor. In general, iPSCs/embryonic stem cells (ESCs) from mammals (e.g., mouse and human) can maintain self-renewal in knock out serum replacement (KSR) base medium. In contrast to mammalian stem cells, our established Okinawa rail, Japanese ptarmigan, and Blakiston's fish owl-derived iPSCs did not show efficient growth in the KSR base medium (Fig. 4a, b). The collective findings indicate that although cell growth does not strongly depend on the presence of LIF and FGF, our established iPSCs derived from Okinawa rail, Japanese ptarmigan, and Blakiston's fish owl require serum to maintain cell growth.

**Cell growth analysis of Janus kinase (JAK) inhibitors, FGFR inhibitors, and protein kinase C (PKC) inhibitors in endangered avian species-derived iPSCs.** Serum contains various growth factors, including inflammatory cytokines and FGF. The JAK-STAT and FGF signaling pathways are critical in maintaining the pluripotency of mammalian iPS/ESCs[9,11,12]. The addition of a JAK inhibitor dramatically inhibited the cell growth and renewability of our established iPSCs (Fig. 4c, d). When FGFR function was inhibited, growth of Okinawa rail and Blakiston's fish owl-derived iPSCs was suppressed (Fig. 4e, f). In contrast to these two avian iPSCs, inhibition of FGFR function did not affect growth Japanese ptarmigan-derived iPSCs, but the colony morphology became jagged and flattened. These findings indicated that FGFR signaling is still important in Japanese ptarmigan-derived iPSCs (Fig. 4e, f). The collective findings demonstrated that JAK and FGFR signaling are important in maintaining the condition of avian-derived iPSCs.

Next, we focused on the inhibition of PKC signaling in Okinawa rail, Japanese ptarmigan, and Blakiston fish owl iPSCs. Takashima et al. reported that human ESCs maintained a naïve-like state in the presence of the Go6983 PKC inhibitor[13]. We hypothesized that the quality of Okinawa rail, Japanese ptarmigan, and Blakiston fish owl iPSCs might improve with the addition of a PKC inhibitor. Indeed, in Japanese ptarmigan and Okinawa rail iPSCs, PKC inhibitors resulted in a positive effect in maintaining the three-dimensional morphology of the iPSC colonies (Fig. 4g). In contrast to Japanese ptarmigan and Okinawa rail iPSCs, Blakiston's fish owl-derived iPSCs displayed arrested cell growth after addition of PKC inhibitor (Fig. 4g). The biological characteristics resulting from additives in the cell culture medium for Okinawa rail, Japanese ptarmigan, and Blakiston's fish owl-derived iPSCs are shown in Fig. 4h. Based on the results from the PKC inhibitor and morphology of iPSC colonies, we suggest that the biological characteristics of Japanese ptarmigan iPSCs might be similar to those of human naïve-like iPSCs among our tested avian-derived cells.

**Resistance against enzymatic digestion in passage procedure and cellular metabolism in iPSC.** To compare the characteristics of these iPSCs, we evaluated their resistance to cell death after enzymatic digestion. Japanese ptarmigan and Okinawa rail-derived iPSCs showed resistance to single-cell enzymatic digestion (Fig. 5a, b). In contrast, Blakiston's fish owl-derived iPSCs did not tolerate enzymatic digestion, and colony growth was strongly suppressed (Fig. 5a, b). Interestingly, when we treated

Blakiston's fish owl-derived iPSCs with a ROCK inhibitor (Y27632), growth of the iPSCs recovered, even when enzymatic digestion was used for cell passage (Fig. 5c, d). Supporting the resistance to cell death after enzymatic digestion, AP activity was increased when the passage procedure was performed by mechanical digestion (Fig. 5c, d and Supplementary Fig. 10). These data show that Blakiston fish owl iPSCs are sensitive to single-cell conditions.

Next, we compared biological differences based on cellular metabolism. In mammals, mitochondrial inner membrane potential and glycolytic metabolism are higher in mouse iPSCs than in human iPSCs[13,14]. We stained mouse, chicken, Okinawa rail, Japanese ptarmigan, and Blakiston fish owl-derived iPSCs with tetramethyl rhodamine methyl ester (TMRE) to evaluate glycolytic metabolism. TMRE visualizes the mitochondrial inner membrane potential using fluorescence intensity. Intense fluorescent signals were observed in mouse iPSCs (positive control) and Japanese ptarmigan-derived iPSCs but not in chicken, Okinawa rail, or Blakiston's fish owl-derived iPSCs (Fig. 5e, f). Mouse, chicken, Okinawa rail, Japanese ptarmigan, and Blakiston's fish owl-derived iPSCs were also stained with MitoTracker, a specific tracker for mitochondria. The fluorescence intensity of MitoTracker in mouse and Japanese ptarmigan iPSCs was higher than the intensity in chicken, Okinawa rail, and Blakiston fish owl-derived iPSCs. These findings indicated that the density of mitochondria in the cytoplasm was relatively high in the former two iPSCs (Fig. 5e, f). We further evaluated the glycolytic metabolism of mouse, chicken, Okinawa rail, Japanese ptarmigan, and Blakiston fish owl-derived iPSCs. Mouse iPSCs generally exhibit higher glucose metabolism than human iPSCs. We cultured cells with 2-deoxyglucose (2DG), a glycolysis inhibitor, to evaluate this activity. When cells exhibit increased glycolysis, they are more tolerant to glycolysis inhibitors. As shown in Fig. 5g, h, Japanese ptarmigan iPSCs maintained AP activity even in the presence of 4 mM 2DG. In contrast, Okinawa rail and Blakiston fish owl iPSCs lost AP activity after 2 mM 2DG treatment. Based on these data, Japanese ptarmigan-derived iPSCs showed elevated glycolysis performance compared to Okinawa rail and Blakiston fish owl-derived iPSCs.

**In vitro differentiation of endangered avian-derived iPSCs.** To evaluate the differentiation of endangered avian iPSCs, we first performed in vitro differentiation using embryoid bodies (EB) (Supplementary Figs. 11a, 12, 13). The Okinawa rail, Japanese ptarmigan, and Blakiston's fish owl iPSC-derived cells differentiated cells to form three germ layer derivatives, as evidenced by the observations of Tuj1, alpha-smooth muscle, and Gata4 positive cells (Supplementary Fig. 11b–d). We observed a dendrite-like morphology in the Japanese ptarmigan iPSCs (Supplementary Fig. 11b). Differentiation of Japanese ptarmigan iPSCs was greatest among the three endangered avian-derived iPSCs.

Next, we evaluated the differentiation of the endangered avian iPSCs to neural cells using all-trans-retinoic acid (ATRA). A dendrite-like morphology of the iPSCs was evident when they were cultured in medium containing ATRA (Supplementary Fig. 11e). On day 21, although Okinawa rail-derived iPSCs allowed observation of dendrite-like morphology, Blakiston's fish owl-derived iPSCs did not show morphological changes to the dendrite-like structure (Supplementary Fig. 11f). These results suggest that differentiation capacity of Okinawa rail iPSCs is higher than that of Blakiston fish owl iPSCs. The collective findings indicate the following order of differentiation capacity: the highest is Japanese ptarmigan iPSCs, second is Okinawa rail iPSCs, and third is Blakiston's fish owl iPSCs (Supplementary Fig. 11g).

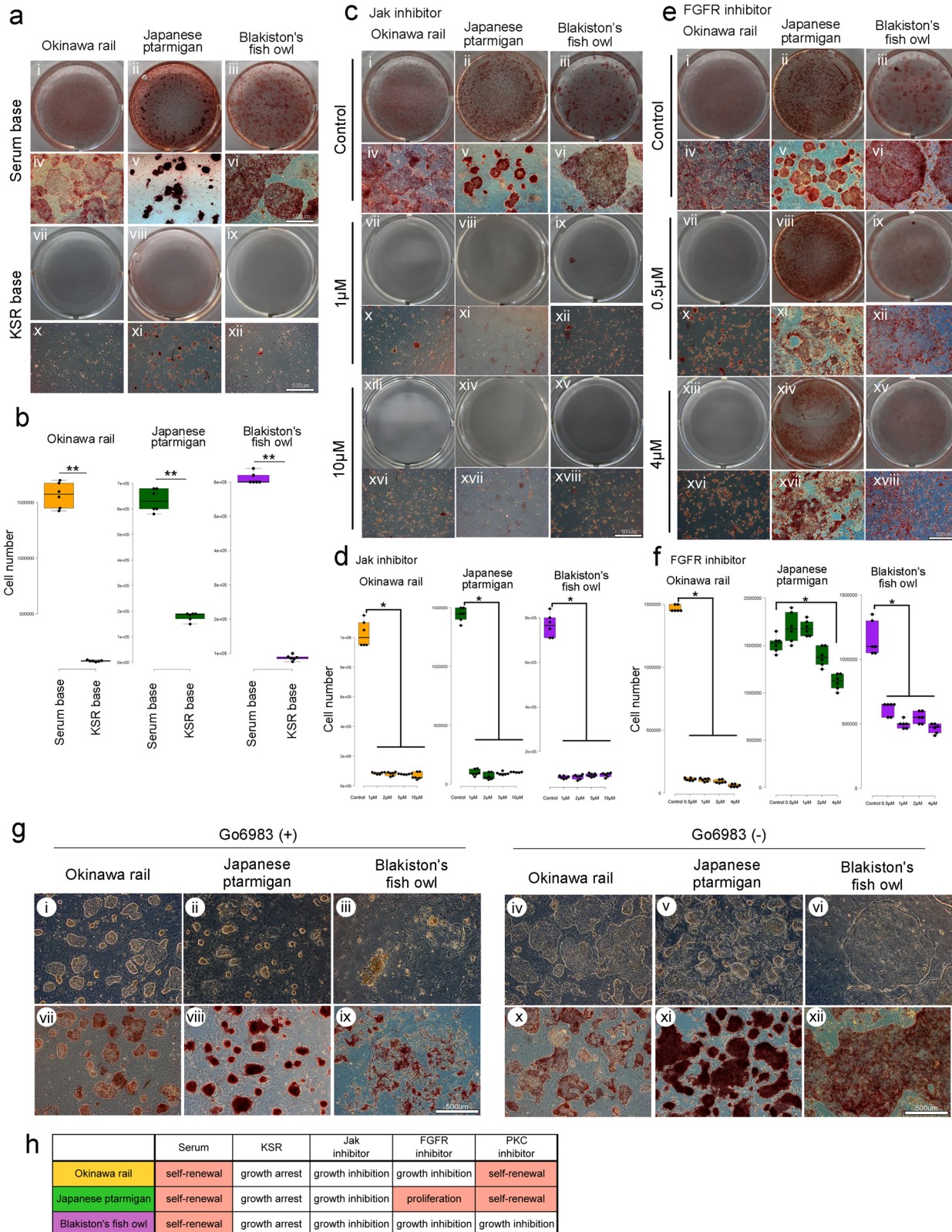

**In vivo differentiation ability of endangered avian-derived iPSCs.** We evaluated the ability of the various iPSCs to differentiate into teratomas in vivo (Supplementary Fig. 14). In contrast to Okinawa rail and Blakiston fish owl iPSCs, Japanese ptarmigan iPSCs rapidly formed tumors. Tumor analyses were performed 6–12 weeks after implantation of Japanese ptarmigan iPSCs and 28–34 weeks after implantation for Okinawa rail and

Blakiston's fish owl iPSCs (Fig. 6a). The teratoma formation ratio of Japanese ptarmigan iPSCs was 100% (17 of 17 mice; Fig. 6b). The tumor formation rate of Okinawa rail and Blakiston's fish owl iPSCs was 70.59 and 42.86%, respectively (Fig. 6b).

Histological analysis revealed that the resulting tumor tissues contained various cell types. Three germ layer structures (ectoderm, mesoderm, and endoderm) were observed in the

**Fig. 4 Okinawa rail, Japanese ptarmigan, and Blakiston's fish owl-derived iPSC growth in knockout serum replacement (KSR) and low molecular inhibitor medium. a, b** Cell growth analysis of Okinawa rail, Japanese ptarmigan, and Blakiston's fish owl-derived iPSCs in serum base medium and knockout serum replacement (KSR) medium. Images show the morphology of the iPSC colonies after alkaline phosphatase staining (**a**). The serum base medium (i–vi) and the KSR base medium (vii–xii) represent. Okinawa rails (i, iv, vii, and x), Japanese ptarmigans (ii, v, viii, and xi), and Blakiston's fish owls (iii, vi, ix, and xii) shows. The bars represent 500 μm. Cell numbers of Okinawa rail, Japanese ptarmigan, and Blakiston's fish owl iPSCs in serum-based and KSR-based media (**b**). Yellow bars represent Okinawa rail, green bars represent Japanese ptarmigans, and purple bars represent Blakiston's fish owl. Centerlines of box plots indicate medians; box limits indicate the 25th and 75th percentiles. n = 6. **P < 0.01. **c** Alkaline phosphatase staining of Okinawa rail, Japanese ptarmigan, and Blakiston's fish owl-derived iPSCs in medium containing JAK inhibitor. Those panels show the control medium (i–vi), medium containing 1 μM JAK inhibitor (vii–xii), and medium containing 10 μM JAK inhibitor (xiii–xviii). Okinawa rail (i, iv, vii, x, xiii, and xvi), Japanese ptarmigans (ii, v, viii, xi, xiv, and xvii), and Blakiston's fish owl (iii, vi, ix, xii, xv, and xviii) -derived iPSCs shows. The bars represent 500 μm. **d** Counting of Okinawa rail, Japanese ptarmigan, and Blakiston's fish owl-derived iPSCs in medium containing JAK inhibitor. Yellow bars show Okinawa rail, green bars show Japanese ptarmigans, and purple bars show Blakiston's fish owl-derived iPSCs. Centerlines of box plots indicate medians; box limits indicate the 25th and 75th percentiles. n = 6. *P < 0.05. **e** Alkaline phosphatase staining of Okinawa rail, Japanese ptarmigan, and Blakiston's fish owl-derived iPSCs in medium containing FGFR inhibitor. Those panels show the control medium (i–vi), medium containing 1 μM JAK inhibitor (vii–xii), and medium containing 10 μM JAK inhibitor (xiii–xviii). Okinawa rail (i, iv, vii, x, xiii, and xvi), Japanese ptarmigans (ii, v, viii, xi, xiv, and xvii), and Blakiston's fish owl (iii, vi, ix, xii, xv, and xviii) -derived iPSCs shows. The bars represent 500 μm. **f** Counting of Okinawa rail, Japanese ptarmigan, and Blakiston's fish owl-derived iPSCs in medium containing FGFR inhibitor. Centerlines of box plots indicate medians; box limits indicate the 25th and 75th percentiles. n = 6. *P < 0.05. **g** Cellular morphology of Okinawa rail, Japanese ptarmigan, and Blakiston's fish owl-derived iPSCs in medium containing Go6983 (PKC inhibitor) and medium lacking Go6983. Bright-field images (i–vi) and alkaline phosphatase (AP) staining images (vii–xii) show. The medium containing Go6983 additive (i–iii and vii–ix) and the medium lacking Go6983 minus (iv–vi and x–xii) show. The Okinawa rail (i, iv, vii, and x), Japanese ptarmigans (ii, v, viii, and xi), and Blakiston's fish owl (iii, vi, ix, and xii) shows. The bars represent 500 μm. **h** Summary of serum and signal dependency.

tumor tissues. In more detail, these included neural tube-like (ectoderm), adipocyte-like (mesoderm), fibroblast-like (mesoderm), chondroblast-like (mesoderm), osteoblast-like (mesoderm), smooth muscle-like (mesoderm), epithelial-like (endoderm), and mucous gland-like (endoderm) (Fig. 6c–e).

Tumor tissue sections were reacted with anti- Tuj1, alpha-smooth muscle, and Gata4 antibodies to examine Okinawa rail, Japanese ptarmigan, and Blakiston's fish owl-derived tumor tissues (Fig. 7a–c). The examination confirmed the histological evidence for the in vivo differentiation of these iPSCs allowed into the three germ layers. Based on in vivo differentiation results, we concluded that our established Okinawa rail, Japanese ptarmigan, and Blakiston's fish owl iPSCs could differentiate into the three germ layers.

**Injection of Japanese ptarmigan-derived iPSCs into chicken embryo.** To test the development and morphogenesis of Japanese ptarmigan iPSCs, these cells were injected into a stage X chicken embryo after fluorescent labeling with a cell tracker (Fig. 8a, b and Supplementary Fig. 15). At 72 h after the injection, cells positive for green fluorescence protein were evident in a small portion of chicken embryos (Fig. 8c and Supplementary Fig. 15). Seventy-two hours following the injection of Japanese ptarmigan iPSCs into stage X chicken embryos, cells in three of 46 embryos displayed green fluorescence. In addition, histological analysis of the embryos was performed to detect the contribution of Japanese ptarmigan-derived iPSCs. The antibody that was used targeted the hygromycin resistance gene product in the reprogramming vector. Strong reactivity with the original Japanese ptarmigan iPSCs was observed (Fig. 8d) and allowed the detection of the contribution of iPSCs to chicken embryos (Fig. 8e).

In addition to the histological analysis, a reprogramming cassette was detected in the chimeric genome. Eleven Japanese ptarmigan iPSC-injected chicken embryos were collected on day five. Of these, six displayed amplification products from the reprogramming vector in genomic PCR analysis (Fig. 8f and Supplementary Fig. 16). We also evaluated the amplification detected by real-time PCR analysis using a fluorescence probe. Amplification of the reprogramming vector sequence was evident in six of the 11 embryos (Fig. 8g, h). The results supported the conclusion that Japanese ptarmigan iPSCs contributed to six of 11 chicken embryos at day 5.

**Transcriptome analysis of avian iPSCs.** To comprehensively compare the expression patterns of the whole genome, RNA-seq analyses of three endangered avian iPSCs, chicken iPSCs, and stage X chicken embryo cells were performed. The sequencing workflow is shown in Fig. 9a. After removing the adapter sequences, we mapped them to the chicken genome. The number of obtained sequence reads was at least 40 M reads, which was sufficient to describe the whole transcriptome (Fig. 9b). In this study, the mapping ratios of all cells exceeded 65% (Fig. 9c). Processing of the data using two-dimensional principle component analysis (PCA) revealed that in stage X chicken embryos, the distance to iPSCs was smaller than that to fibroblasts in the four avian species (Fig. 9d). Dendrogram analysis using a heat map analysis was also performed (Fig. 9e). The data of seven reprogramming factors from chicken iPSCs were compared with previous data of chicken ESCs[15]. These cell distances were much smaller than those of chicken fibroblasts and stage X cells; therefore, the iPSCs derived in the present study were similar to chicken ESCs (Fig. 9f). Hwang et al. reported the transcriptional profile of chicken developmental stage cells[16]. We compared chicken embryo stage X cells and iPSCs with the chicken development stage using these data (Fig. 9g). Our chicken embryo stage X cells showed similar expression profile of stage X compared with chicken fibroblasts and chicken iPSCs. We concluded that our established avian iPSCs were closer to chicken embryo stage X cells than to fibroblasts.

**Establishment of Japanese golden eagle iPSCs.** We attempted to establish Japanese golden eagle iPSCs from pinfeather-derived somatic cells. First, we tried to establish Japanese golden eagle iPSCs with the PB-TAD-7F reprogramming vector (Supplementary Fig. 17a–d). Since iPSC-derived teratomas did not form in severe combined immunodeficiency (SCID) mice (Supplementary Fig. 17e), we developed a new reprogramming method to establish Japanese golden eagle iPSCs. In the novel method, to reprogram somatic Japanese golden eagle cells into stem cells, we designed the PB-DDR-8F reprogramming vector. This vector contains a PiggyBac transposon backbone and eight different genes that promote reprogramming in mammalian cells. The order of reprogramming genes was *Oct3/4*, *Sox2*, *Klf4*, *c-Myc*, *Klf2*, *Lin28*, *Nanog*, and *Yap* (Fig. 10a). In a previous study

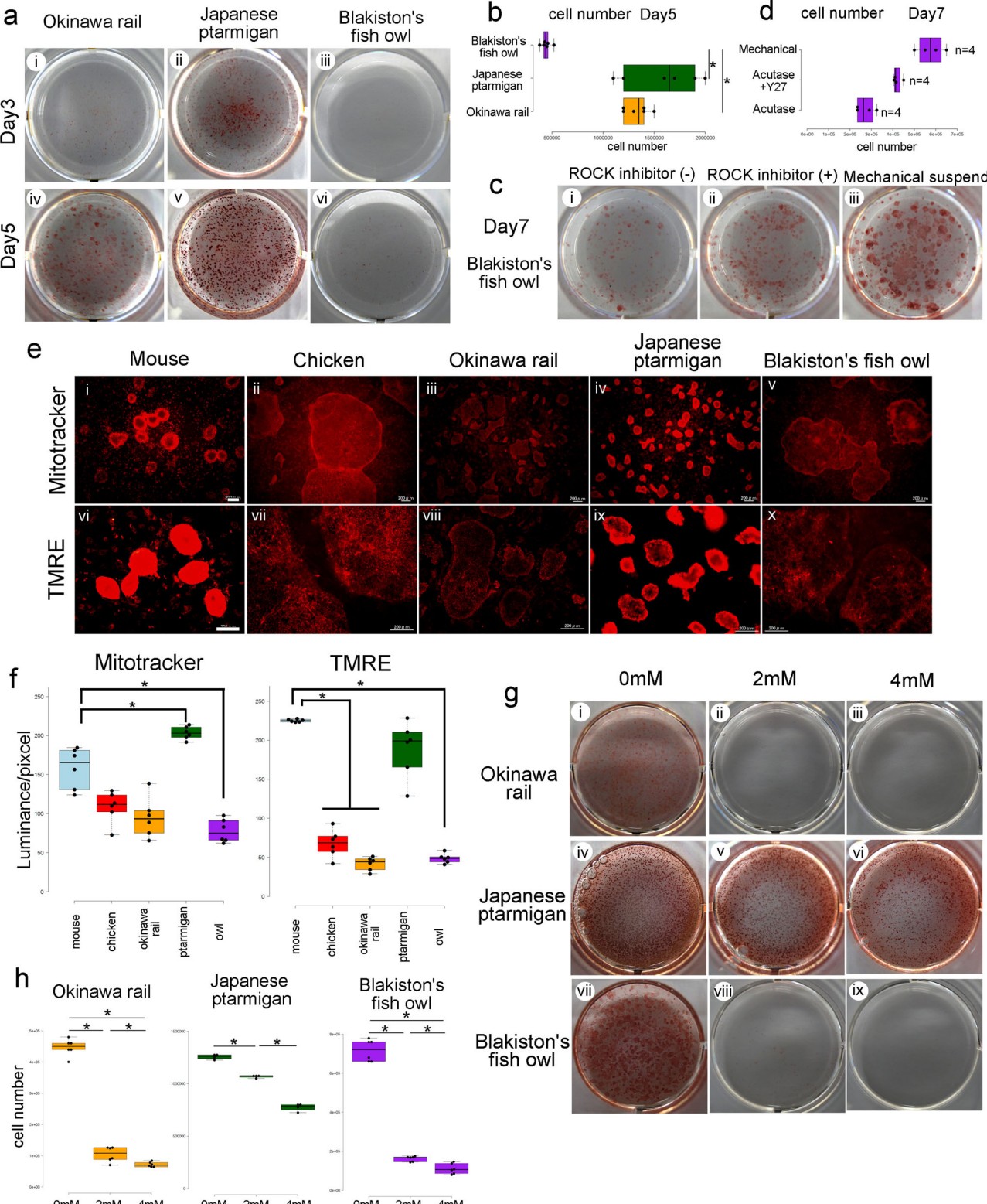

involving human PSCs, the expression of *Yap/Taz* genes improved stem cell quality[17]. Therefore, we hypothesized that *Yap/Taz* expression could improve stem cell quality in avian species. Furthermore, to enhance the transcriptional activity of *Oct3/4*, we introduced the amino acid sequence DDR-DDR-DDR at the N-terminus of *Oct3/4* (Fig. 10a and Supplementary Fig. 18).

We attempted to establish Japanese golden eagle iPSCs as detailed in Fig. 10b. After introducing the PB-DDR-8F

reprogramming vector into Japanese golden eagle somatic cells, many (at least 20) primary colonies were obtained. Eight of these colonies were picked and eight iPSC clones were established (Fig. 10c). We focused on analyzing three Japanese golden eagle iPSC clones, and these clones were positive for the genomic insertion of the corresponding expression cassette (Fig. 10d). Flow cytometry did not detect a shift in the histograms of iPSCs compared to that of the somatic cells (Fig. 10e). Karyotype

**Fig. 5 Evaluation of stem cell qualities of Okinawa rail, Japanese ptarmigan, and Blakiston's fish owl iPSCs. a** Colony formation image of Okinawa rail, Japanese ptarmigan, and Blakiston's fish owl iPSCs after digestion with Accutase on days 3 and 5. The day 3 (i–iii), and the day 5 (iv–vi) show. Okinawa rail iPSCs (i and iv), Japanese ptarmigan iPSCs (ii and v), Blakiston's fish owl (iii and vi) iPSCs. **b** Cell numbers of the three endangered avian-derived iPSCs after digestion with Accutase on day 5. Yellow bar: Okinawa rail iPSCs; green bar: Japanese ptarmigan iPSCs; and purple bar: Blakiston's fish owl iPSCs. $n = 6$. $*P < 0.05$. Centerlines of box plots indicate medians; box limits indicate the 25th and 75th percentiles. **c** Alkaline phosphatase staining of Blakiston's fish owl iPSCs on day 7. This image show the iPSC colonies of digestion with Accutase and Y27632 minus (i), digestion with Accutase and Y27632 plus (ii), and mechanical passage (iii). **d** Enumeration of cell number of Blakiston's fish owl iPSCs on day 7. Centerlines of box plots indicate medians; box limits indicate the 25th and 75th percentiles. $n = 4$. $*P < 0.05$. **e** Mitochondrial staining of iPSCs. MitoTracker staining (i–v) and TMRE staining (vi–x) of mouse (i and vi), chicken (ii and vii), Okinawa rail (iii and viii), Japanese ptarmigan (iv and ix), and Blakiston's fish owl (v and x) shows. The bars represent 200 μm. **f** Detection of MitoTracker or TMRE density in iPSCs. Blue bar: mouse iPSCs; red bar: chick iPSCs; yellow bar: Okinawa rail iPSCs; green bar: Japanese ptarmigan iPSCs; purple bar: Blakiston fish owl iPSCs. ($n = 6$). Centerlines of box plots indicate medians; box limits indicate the 25th and 75th percentiles.$*P < 0.05$. **g** Alkaline phosphatase-positive colony formation of Okinawa rail, Japanese ptarmigan, and Blakiston's fish owl iPSCs after 2-deoxyglucose (2DG) treatment (0 mM, 2 mM, 4 mM). Okinawa rail iPSCs (i–iii), Japanese ptarmigan iPSCs (iv–vi), and Blakiston fish owl iPSCs (vii–ix) show. 0 mM 2DG (i, iv, and vii), 2 mM 2DG medium (ii, v, and viii), and 4 mM 2DG medium (iii, vi, and ix). **h** Cell counting of the Okinawa rail, Japanese ptarmigan, and Blakiston's fish owl iPSCs treated with 2DG. Centerlines of box plots indicate medians; box limits indicate the 25th and 75th percentiles. $n = 6$ (Okinawa rail and Blakiston's fish owl), $n = 4$ (Japanese ptarmigan). $*P < 0.05$.

analysis did not reveal any difference in the number of chromosomes between somatic cells and iPSCs (Fig. 10f). Therefore, we concluded that Japanese golden eagle iPSCs did not have polyploidy or chromosomal abnormalities.

Japanese golden eagle iPSC colonies were positive for AP activity (Fig. 10g). Furthermore, these iPSCs positively stained for SSEA-1 and -3 (Fig. 10h), indicating their expression of multiple pluripotency markers. The iPSCs strongly expressed numerous pluripotency-related genes, such as *Sox3-like*, *Nanog-like*, *Lin28a*, *Lin28b*, *TERT*, *CDH1*, *Sall4*, and *Esrrb* (Fig. 10i). Although *POU5* (mammalian *Oct3/4*) is one of the main pluripotency genes in avian species, we could not find genomic information concerning *POU5* in the golden eagle genome database. Based on the collective data, the Japanese golden eagle cells fit the criteria for PSCs.

**Analysis of differentiation of Japanese golden eagle iPSCs.** We evaluated the differentiation ability based on teratoma formation in SCID immunodeficient mice. The Japanese golden eagle iPSCs formed large teratomas in the SCID mouse testes (Supplementary Figs. 19, 20). Tumors formed in six of eight testes after implantation of the iPSCs (Supplementary Fig. 19). Histological analysis of the teratomas revealed cellular differentiation into the ectoderm (neural tube-like structures), mesoderm (fibroblasts and fat), and endoderm (epithelial-like structures) layers (Supplementary Fig. 20a–f). The results demonstrated that that Japanese golden eagle iPSCs can differentiate into the three germ layers.

We also assessed the in vitro differentiation of the established Japanese golden eagle iPSCs. EB formation from these iPSCs was induced in vitro (Supplementary Fig. 20g). The expressions of Tuj1 (ectoderm marker), alpha-SMA (mesoderm marker), and Gata4 (endoderm marker) were detected in differentiated cells (Supplementary Fig. 20h–j). The results also indicated that Japanese golden eagle iPSCs could differentiate into three germ layers in vitro.

Finally, we analyzed the differentiation of Japanese golden eagle iPSCs into neural cells in the presence of ATRA. After the addition of ATRA to the culture medium, the cellular morphology changed to a dendrite-like morphology (Supplementary Fig. 20k). The dendrite-like cells stained positive for the Tuj1 dendritic cell marker (Supplementary Fig. 20i). We next evaluated the differentiation process by comparing the gene expression of iPSCs, differentiated cells from EB (ATRA−), and differentiated cells from EB in the presence of ATRA (ATRA+). Expressions of the neural cell markers Nestin and Pax6 were higher in ATRA+ cells than in ATRA− cells (Supplementary Fig. 20m). Thus, treatment with ATRA enhanced neural differentiation of Japanese golden eagle iPSCs. The collective findings demonstrated the ability of these iPSCs to differentiate into various cell types in vitro.

## Discussion

Although mammalian iPSCs have been established from various species, including rabbit, common marmoset, and pig, avian-derived iPSCs have been limited to a few species, such as chicks and quails[2,3,5,18–21]. Therefore, the establishment of avian-derived iPSCs is considered to be much more complicated than that of mammalian-derived iPSCs. To generate endangered avian iPSCs, we performed numerous trial-and-error experiments. For example, established endangered avian iPSCs with six factors (normal *Oct3/4*, *Sox2*, *Klf4*, *c-Myc*, *Nanog*, and *Lin28*) in the PB-6F reprogramming vector (the vector information is provided in a previous study), while reprogrammed cells are unable to maintain long-term passage[22]. We would like to establish the iPSCs with PB-R6F reprogramming vector, Blakiston's fish owl-derived primary colonies did not appear in our study (Supplementary Fig. 2). Furthermore, for the establishment of golden eagles iPSCs, we further improved the reprogramming method. Based on these trials and error experiments, we succeeded in establishing iPSCs from three endangered avian species (Okinawa rail, Japanese ptarmigan, and Blakiston's fish owl) using Oct3/4 with enhanced transcriptional activity and an increased number of reprogramming factors. Furthermore, we established Japanese golden eagle iPSCs using a PB-DDR-8F reprogramming vector. To the best of our knowledge, this is the first report of the establishment of iPSCs from endangered avian species. Successful establishment of iPSCs can control their differentiation into various cells.

Although we used identical reprogramming methods for Okinawa rail, Japanese ptarmigan, and Blakiston's fish owl cells, we observed species differences in the biological characteristics of the established iPSCs (Fig. 10j). In the generation of iPSCs, reprogramming gene homologies are critical. Therefore, we initially hypothesized that polymorphisms in the amino acid sequence of reprogramming genes might explain the biological differences among Okinawa rail, Japanese ptarmigan, and Blakiston's fish owl iPSCs. Comparison of the amino acid sequences of the three endangered avian species with mouse sequence did not reveal a significant difference in the DNA-binding domain sequence between the sequences (*POU5*: 65–67%, *Sox2*: 97%, *Klf4*: 90–97%, *c-Myc*: 89%, *Nanog*: 60–65%, *Lin28a*: 91–92%, *Klf2*: 87–95%) (Supplementary Methods, Supplementary Figs. 21–24). Based on these results, differences in the primary sequences of reprogramming-related genes do not explain the biological outcomes among the iPSCs of these species.

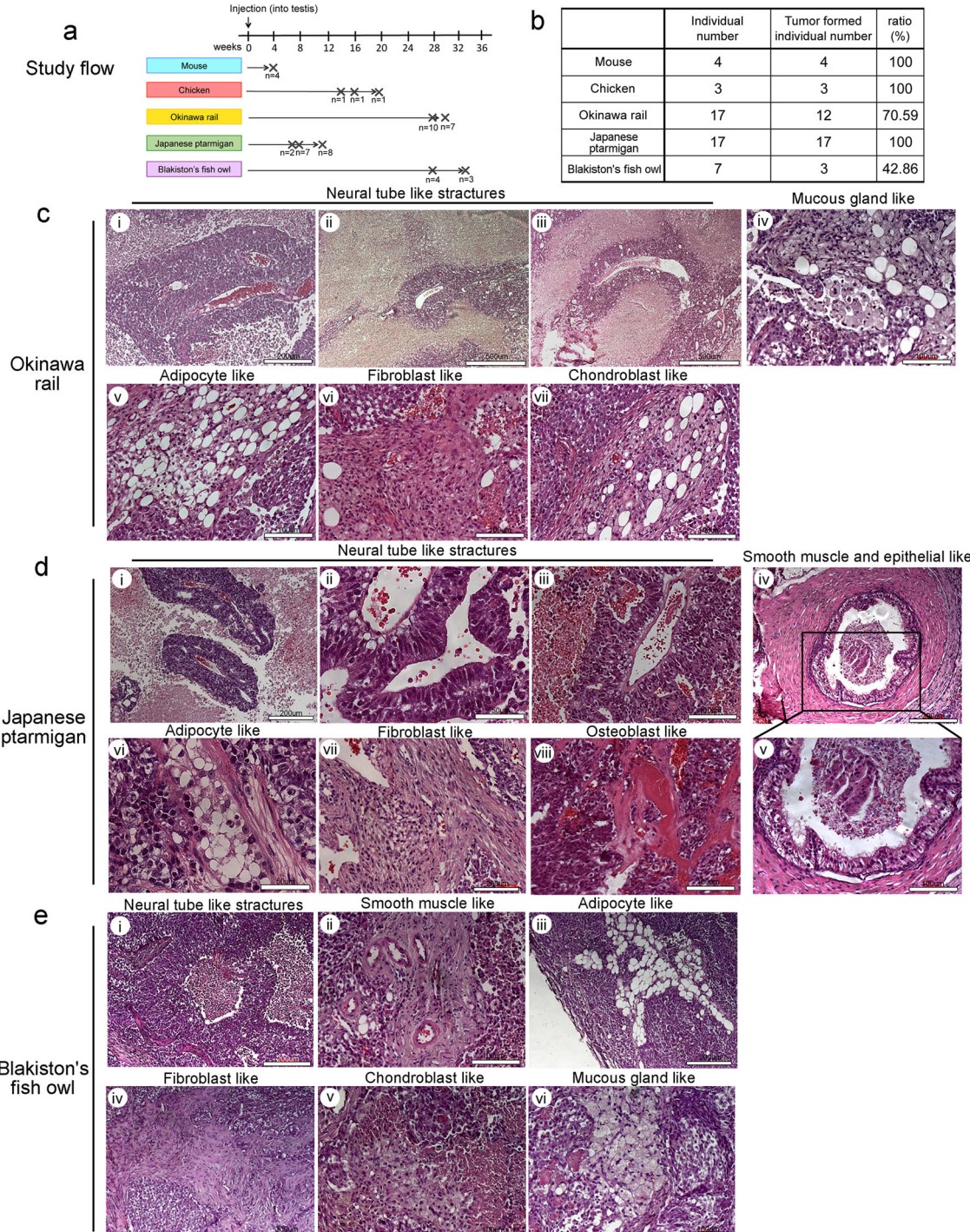

**Fig. 6 In vivo differentiation of established mouse, chicken, Okinawa rail, Japanese ptarmigan, and Blakiston's fish owl-derived iPSCs. a** Tumors that formed in testis tissue of SCID mice. **b** Tumor formation ratio. Histological analysis of Okinawa rail (**c**), Japanese ptarmigan (**d**), and Blakiston's fish owl (**e**) iPSC-derived teratomas. Various tissues originating from the three germ layers were identified and included neural tube-like structure (ectoderm) (**c**i, **c**ii, **c**iii, **d**i, **d**ii, **d**iii, and **e**i), adipocyte-like structure (mesoderm) (**c**v, **d**vi, and **e**iii), fibroblast-like structure (mesoderm) (**c**vi, **d**vii, and **e**iv), chondroblast-like structure (mesoderm) (**c**vii, and **e**v), osteoblast-like structure (mesoderm) (**d**viii), smooth muscle-like structure (mesoderm) (**d**iv, and **d**v), mucous gland-like structure (endoderm) (**c**iv, and **e**vi), and epithelial-like structure (endoderm) (**d**iv, and **d**v). The tissues were then stained with hematoxylin and eosin. Bars indicate 50–500 µm.

In this study, three clones of Okinawa rail iPSCs showed a reduction in the GFP expression during five passages (passages 5 to 9), while three clones of ptarmigan iPSCs maintained the GFP expression (passages 6 to 10) (Supplementary Fig. 25). In addition to their sequential passage, we confirmed the endogenous pluripotency-related gene expression in passages 3 and 10 with real-time PCR. Although Okinawa rail iPSCs did not maintain the exogenous gene expression, major pluripotency-related genes, such as *POU5*, *Nanog* and *Lin28* maintained the higher-level expression during passages 3 and 10. These results indicated that expression of major pluripotency marker genes continues regardless of the expression levels of exogenous genes. We

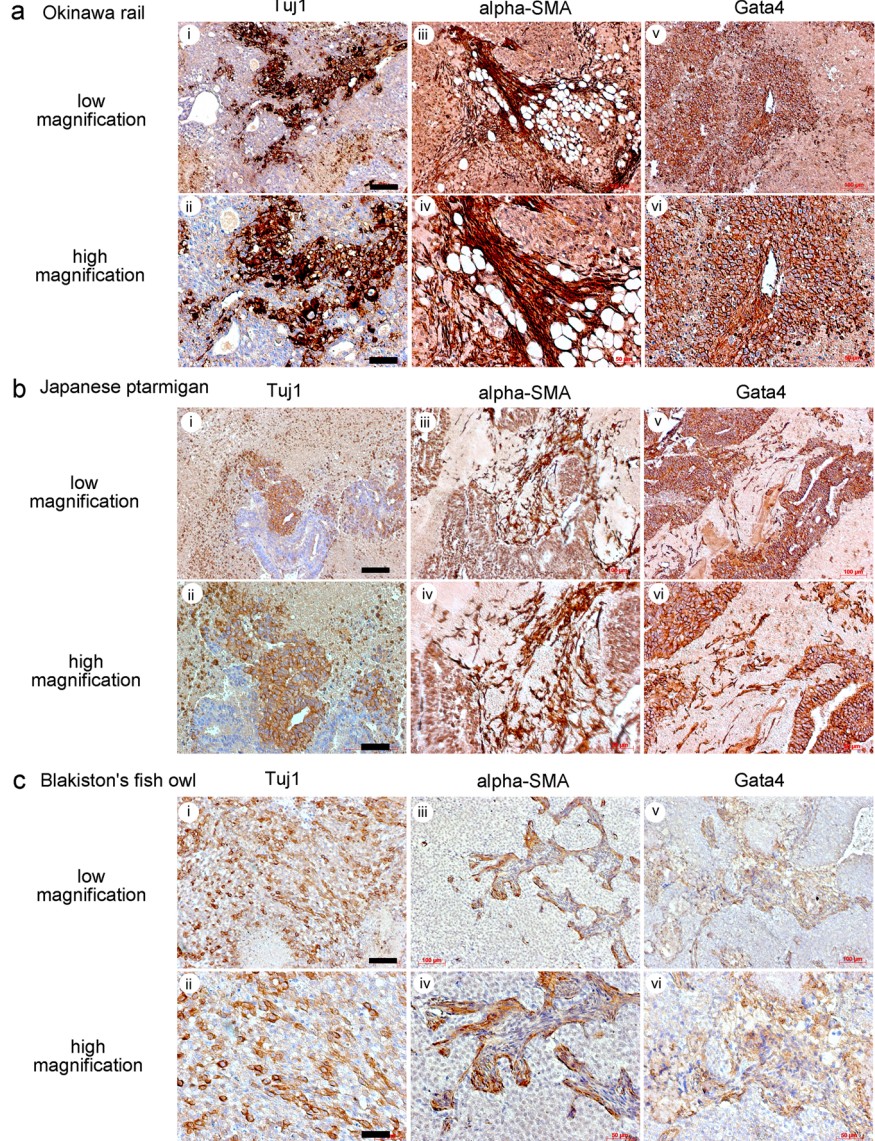

**Fig. 7 Histological analysis with immunological staining of Okinawa rail, Japanese ptarmigan, and Blakiston's fish owl iPSCs derived teratomas.**
Histological analysis with Tuj1, alpha-smooth muscle (SMA), Gata4 antibody. Immunological staining of Okinawa rail (**a**), Japanese ptarmigan (**b**), and Blakiston fish owl (**c**) iPSC-derived teratomas. Those panels shows low magnification (i, iii, v) and high magnification (ii, iv, vi). Histological analysis with Tuj1 antibody (i, ii), histological analysis with alpha-smooth muscle (SMA) antibody (iii, iv), histological analysis with Gata4 antibody (v, vi). Bars indicate 100 μm (i, iii, v) or 50 μm (ii, iv, vi).

therefore considered that different cellular characteristics among three endangered avian are species differences, not exogenous gene expression levels.

Although we did not clearly mention the origin of the species differences, species differences in the promoter or non-coding genomic regions may be responsible. Our data on pluripotency provides valuable information for avian stem cell scientists and conservation biologists.

## Methods

**Animal experiments**. Teratoma formation experiments were performed at Iwate University. All surgical procedures and animal husbandry were performed in accordance with the international guidelines of the Animal Experiments of Iwate University and were approved by the university's Animal Research Committee (approval number A201734).

Chicken embryonic fibroblasts (Rhode Island Red) were obtained from a primary culture of chicken embryonic tissue provided by Prof. Atsushi Tajima, Tsukuba University. Chicken culture cells were obtained from chicken embryos, and the acquisition of these cells did not require approval. Mouse embryonic

fibroblasts (CF-1 strain) were purchased from a manufacturer (CMPMEFCFL; DS Pharma Biomedical, Osaka, Japan). Approval was not required to obtain these cells.

Somatic cells were obtained from wild animals (ex., Okinawa rail). The sampling details described below do not include the exact location of sampling to protect against poaching.

Fibroblast cells from Okinawa rail and Japanese ptarmigan were obtained from dead animals, such as those killed by vehicles (Fig. 1A and Supplementary Fig. 1). Approval was not required to obtain these samples.

Dead Okinawa rail were found on May 21, 2008, by the Okinawa Wildlife Federation, a nonprofit organization that focuses on the conservation of wild animals in the Okinawa area in the southwest region of Japan. The organization has permission from the Japanese Ministry of the Environment (MOE) to handle and perform first aid activities on endangered animals. The dead birds were transferred the following day to the National Institute for Environmental Studies (NIES). Primary cell culture was carried out from muscle tissue and skin of the dead birds (NIES ID: 715A).

On July 8, 2004, tissues recovered from dead Japanese ptarmigan (e.g., skin and retina tissues) were also transferred to NIES from Gifu University Department of Veterinary Medicine. Primary cell culture from this tissue was performed (NIES ID: 22A).

Somatic cells from Blakiston's fish owl and Japanese golden eagle were obtained from emerging pinfeathers. Concerning the Blakiston's fish owl, the MOE carries

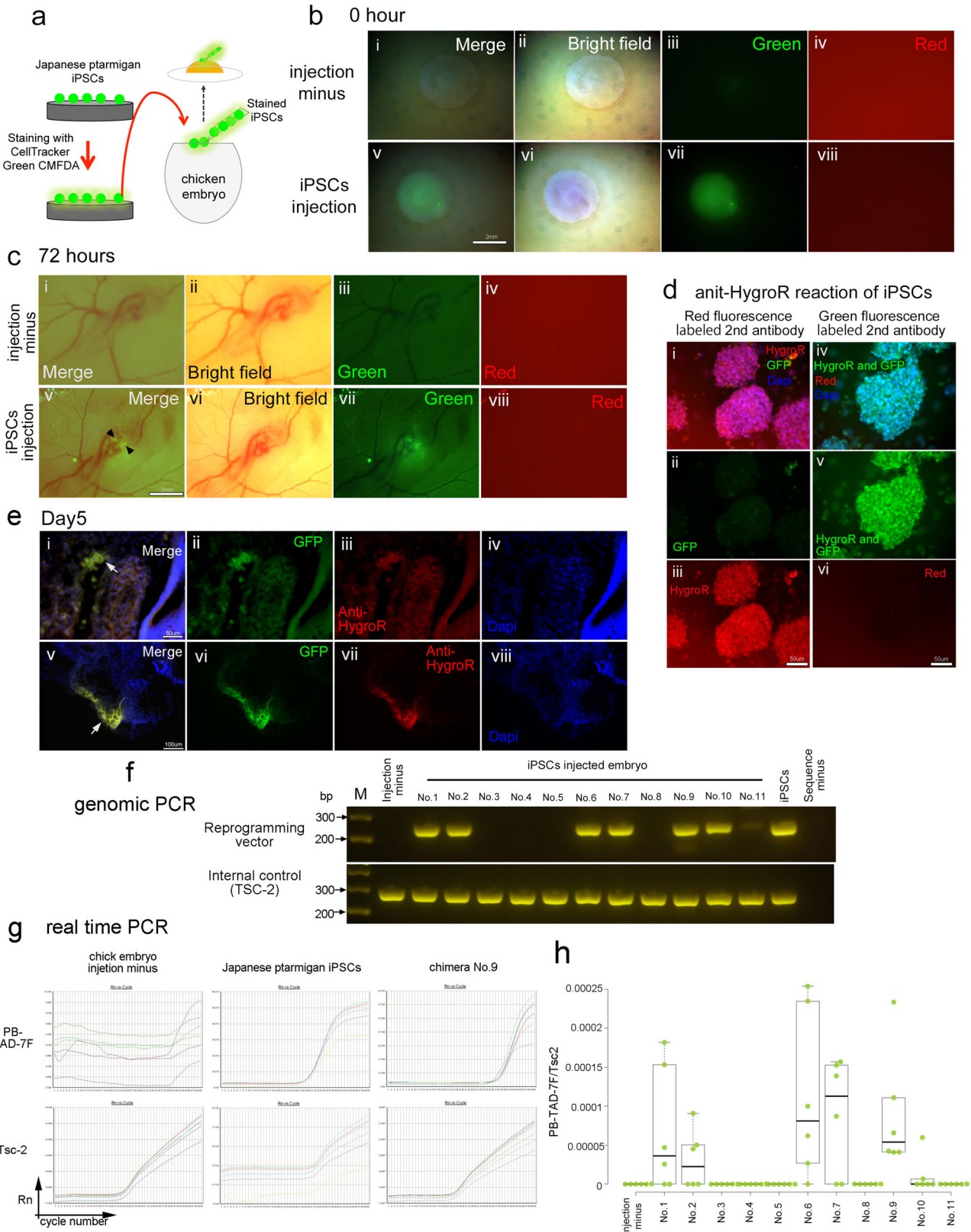

out bird banding, of wild birds with identification tags. The emerging pinfeathers we used had been accidentally release during banding. The banding had been performed by a veterinarian at the Institute for Raptor Biomedicine Japan (IRBJ) in the Hokkaido area on June 2, 2006. IRBJ is a private organization that primarily focuses on emergency medicine first aid and care for wild avians in Hokkaido region of Japan. IRBJ is contracted to MOE to handle and administer first aid for endangered animals. The MOE banding ring was 14C0242. Since banding was carried out with the permission of MOE for capturing wildlife, we did not require

the approval to obtain these avian somatic cells. On July 8, 2006, Blakiston's fish owl pinfeathers were transferred to from IRBJ to NIES, where primary cell culture was performed (NIES ID: 215A).

Concerning the Japanese golden eagle, an emerging pinfeather accidentally fell off a bird during blood collection at the Yagiyama Zoo in Sendai, Japan on July 11, 2018. Dr. Yukiko Watanabe, an IRBJ veterinarian, collected the emerging pinfeather. The sample was shipped the following day to NIES where primary cell culture was performed (NIES ID: 5228).

**Fig. 8 Contribution of Japanese ptarmigan-derived iPSCs to chicken embryos. a** iPSCs were injected into chicken blastoderm after fluorescently labeling them with a cell tracer. **b** Interspecific blastoderm was generated after the injection of Japanese ptarmigan-derived iPSCs into chicken blastoderm. Images shows embryos that were not injected (i–iv) and iPSC-injected blastoderm (v–viii). Panels show the merged bright field and green fluorescence images (i, v), bright field images (ii, v), green fluorescence (iii, vii), and red fluorescence (iv, viii). Bars represent 2 mm. **c** The images show the contribution of iPSCs to the chicken embryo post-injection after 72 h. Arrows indicate green signals. Panels show embryos with no injection (i–iv), and iPSC-injected blastoderm (v–viii). Panels show merged bright field and green fluorescence images (i, v), bright fields (ii, vi), green fluorescence (iii, vii), and red fluorescence (iv, viii). Bars represent 2 mm. **d** Reaction of anti-hygromycin antibody with Japanese ptarmigan iPSCs. Staining images of the anti-hygromycin antibody with red fluorescence-labeled secondary antibody (i–iii), staining images of the anti-hygromycin antibody with green fluorescence-labeled secondary antibody (iv–vi). Panels show merged images (i, iv), green fluorescence (ii, v), and red fluorescence (iii, vi). Scale bar represents 50 μm. **e** Detection of GFP, hygromycin resistance gene, and DAPI signals in embryonic cryosections on day 5. Panels show the merged image (i, v), the GFP (ii, vi), the hygromycin resistance gene (iii, vii), and the DAPI (iv, viii) stained images. Bars represent 50 μm (upper) and 100 μm (lower). **f** Detection of reprogramming vector cassette from Japanese ptarmigan-injected chicken embryos on day 5. The electrophoretic image of the detection of the reprogramming vector (PB-TAD-7F) and the detection of the internal control (Tsc-2) shows. **g, h** Detection of reprogramming vector cassette from Japanese ptarmigan-injected chicken embryos on day 5 with real-time PCR. **g** Amplification plots of PB-TAD-7F (reprogramming vector) and Tsc-2 in chicken embryo (injection minus), Japanese ptarmigan iPSCs, and chimera No. 9 (iPSC-injected chicken embryo No. 9). **h** Quantification of PB-TAD-7F in iPSC-injected chicken embryos. Centerlines of box plots indicate medians; box limits indicate the 25th and 75th percentiles. $n = 6$.

In addition to these birds, we obtained somatic cells emerging avian pinfeathers of Steller's sea eagle, white-tail eagle, mountain hawk-eagle, northern goshawk, Taiga bean goose, and Latham's snipe. These samples were provided by IRBJ.

Concerning the Steller's sea eagle, an injured individual was found in Hokkaido on July 11, 2006 (ID: 06-NE-SSE-1). The eagle was transferred to IRBJ. On December 4, 2006, IRBJ veterinarian Dr. Keisuke Saito collected fallen pinfeathers. Primary cell culture was performed at NIES on December 8, 2006 (NIES ID: 369A).

Concerning the white-tailed eagle, an injured individual was found in Hokkaido, Japan, on July 12, 2007 (ID: 07-NE-WTE-4). The bird was transferred to IRBJ the same day for emergency treatment. On January 15, 2008, Dr. Saito collected fallen pinfeathers. Primary cell culture was performed on January 18, 2008 at NIES (NIES ID: 492A).

Concerning the mountain hawk-eagle, an injured individual was found in the Hokkaido area on August 10, 2008 (ID: 08-Tokachi-HHE-2). The bird was transferred to IRBJ the same day. The bird was treated by an IRBJ veterinarian, but died on September 8, 2008. Emerging pinfeathers were collected from the dead bird by Dr. Saito. Primary cell culture was performed on September 11, 2008 at NIES (NIES ID: 847A).

Concerning the Northern Goshawk, IRBJ accepted an injured bird for treatment on June 12, 2006. Following treatment and recovery, the bird was released into the wild in the Hokkaido area on August 1, 2006. During the treatment (July 4, 2006), Dr. Saito collected fallen pinfeathers. The primary cell culture was performed at NIES on July 6, 2006 (NIES ID: 222A).

Concerning the Taiga bean geese, an injured individual was found in Hokkaido on September 15, 2016 (ID: 13B8005). The injured bird was transferred to IRBJ the same day for emergency treatment. On September 16, 2016, IRBJ veterinarian Dr. Yukiko Watanabe collected fallen emerging pinfeathers. Primary cell culture was performed on September 20, 2016 (NIES ID: 4420A).

Finally, concerning the Latham's snipe, fallen pinfeathers were collected during MOE approved bird banding performed on September 17, 2006, by Dr. Saito. Dr. Saito also collected fallen emerging pinfeathers (ID: 6A22598). The samples were transferred to NIES on September 20, 2006, for primary cell culture (NIES ID: 338A).

All records are available at NIES.

**Cell culture and preservation**. Okinawa rail, Japanese ptarmigan, and Blakiston's fish owl-derived fibroblasts were preserved in liquid nitrogen for 8–12 years (Fig. 1f). The preservation solution contained 90% fetal bovine serum (FBS) and 10% dimethyl sulfoxide. Cells were preserved at a cell density of $1 \times 10^6$–$4 \times 10^6$ cell/mL. During the freezing period, the cells were maintained at minus The cells were frozen at a temperature of $-135\,°C$. Japanese golden eagle fibroblasts were used without freezing.

Avian-derived fibroblasts were cultured with Kuwana's modified avian culture medium-1 (KAv-1), which is based on alpha-MEM containing 5% FBS and 5% chicken serum[23]. Mouse embryonic fibroblasts were cultured in Dulbecco's modified Eagle's medium (DMEM) containing 10% FBS and 1% antibiotic–antimycotic mixed stock solution (161–23181; Wako Pure Chemical Industries, Osaka, Japan). All avian and mouse cells were cultured at 37 °C under 5% $CO_2$.

**Reprogramming vector**. We chemically synthesized an expression cassette that included seven reprogramming factors (MyoD-derived transactivation domain-linked *Oct3/4*, *Sox2*, *Klf4*, *c-Myc*, *Klf2*, *Lin28*, and *Nanog*; all genes derived from mice). The self-cleaving 2A peptide was inserted at the junction of the coding region (Fig. 1g). We transferred the complementary DNA (cDNA) insert from the shuttle vector to the PiggyBac transposon vector containing green fluorescent protein (PB-CAG-GFP). Although the original transposon vector drive the

expression of cDNA with the elongation factor-1 (EF1) promoter (PJ547-17; DNA 2.0, Menlo Park, CA, USA), we replaced the EF1 promoter to CAG promoter in our previous study[22,24]. The reprogramming vector was designated PB-TAD-7F (Fig. 1g).

In addition to the PB-TAD-7F reprogramming vector, we used the PB-DDR-8F reprogramming vector to establish Japanese golden eagle iPSCs. The complete coding sequence of DDR-8F (DDR-*Oct3/4*, *Sox2*, *Klf4*, *c-Myc*, *Klf2*, *Nanog*, *Lin28*, and *Yap*) was chemically synthesized. The expression cassettes containing the eight reprogramming factors were excised from the shuttle vector using restriction enzymes. The cDNA fragments were transferred to the PB-CAG-GFP PiggyBac transposon-based vector[22,24]. Detailed information regarding the PB-DDR-8F reprogramming vectors is shown in Fig. 10a.

**Establishment of iPSCs**. We transfected PB-R6F or PB-TAD-7F reprogramming vectors into mouse, chicken, Okinawa rail, Japanese ptarmigan, and Blakiston's fish owl-derived fibroblasts using Lipofectamine 2000 transfection reagent (Thermo Fisher Scientific, Waltham, MA, USA). After hygromycin selection (Wako Pure Chemical Industries), the cells were reseeded onto a mouse embryonic fibroblast (MEF) feeder layer. On days 14–32, we picked primary iPSC-like colonies and seeded them on new MEF feeder cell plates. The detailed protocol is shown in Fig. 1h.

To establish Japanese golden eagle-derived iPSCs, we transduced PB-TAD-7F or PB-DDR-8F reprogramming vectors into Japanese golden eagle pinfeather-derived somatic cells. Transfection was performed using Lipofectamine 2000 transduction reagent (11668019; Thermo Fisher Scientific) according to the manufacturer's instructions. After hygromycin selection (Wako Pure Chemical Industries), cells were seeded onto feeder culture plates. The golden eagle iPSCs were cultured in KAv-1-based medium[5].

The medium used to establish avian iPSCs was supplemented with 1000 × human Leukemia Inhibitory Factor (LIF) (125–05603; Wako Pure Chemical Industries), 4.0 ng/ml basic FGF (064–04541; Wako Pure Chemical Industries), 0.75 μM CHIR99021 glycogen synthase kinase-3 inhibitor (034–23103; Wako Pure Chemical Industries), 0.25 μM PD0325901 mitogen-activated protein kinase inhibitor (163–24001; Wako Pure Chemical Industries). In addition to those supplements, 0.25 μM thiazovivin (202–18011; Wako Pure Chemical Industries) was added in the media used to generate Okinawa rail, Japanese ptarmigan, Blakiston's fish owl, and chicken iPSCs. In the medium used to generate mouse iPSCs, we added 1000 × LIF, 0.75 μM CHIR99021, and 0.25 μM PD0325901.

**iPSC culture conditions**. Two types of cell culture media were used: KAv-1 for avian iPSCs and DMEM for mouse iPSCs. The composition of KAv-1 for avian iPSCs was as follows: alpha-MEM containing 5% FBS and 5% chicken serum 1% antibiotic–antimycotic mixed solution, 1% nonessential amino acids (Wako Pure Chemical Industries), and 2 mM glutamic acid was added (Nacalai Tesque, Kyoto, Japan). The composition of DMEM for mouse was as follows: DMEM supplemented with 15% SSR, 0.22 mM 2-mercaptoethanol (21438–82, Nacalai Tesque), 1% antibiotic–antimycotic mixed solution, 1% nonessential amino acids[5,22]. As a supplement to the iPSC medium, we used 1000 × human LIF (125–05603; Wako Pure Chemical Industries), 4.0 ng/ml basic FGF (064–04541; Wako Pure Chemical Industries), 0.75 μM CHIR99021 (034–23103; Wako Pure Chemical Industries), 0.25 μM PD0325901 (163–24001; Wako Pure Chemical Industries) for the media used to culture Okinawa rail, Japanese ptarmigan, Blakiston's fish owl, Japanese golden eagle, and chicken-derived iPSCs. The supplements for media used to culture Okinawa rail and Japanese ptarmigan-derived iPSCs included 2.5 μM Gö6983 (074–06443, Wako Pure Chemical Industries). To analyze the cellular characteristics, we focused on the Janus kinase (JAK), FGF, ROCK, and glycolytic pathways, since the dependency of these pathways can indicate differences in

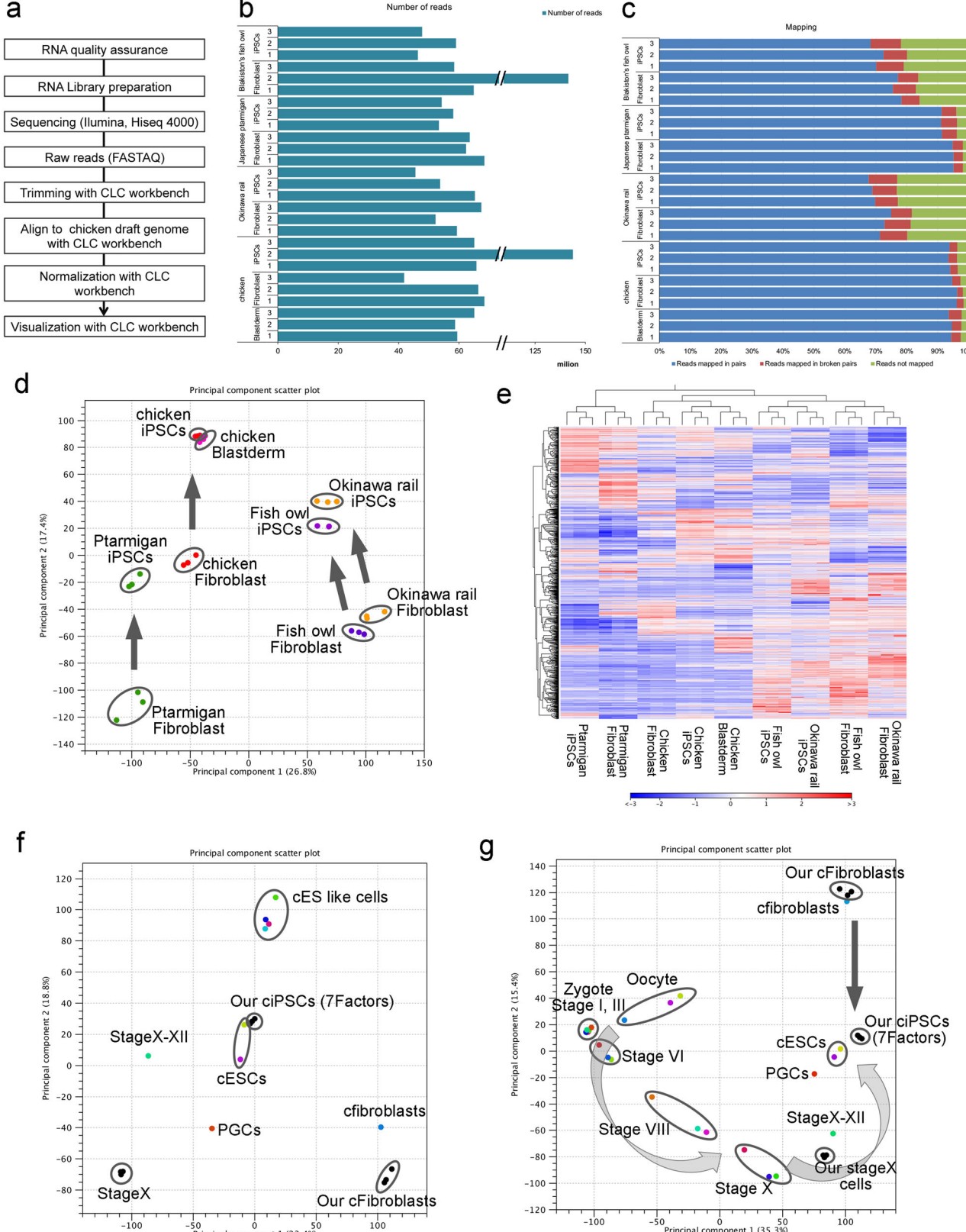

**Fig. 9 Global gene expression profile of engendered avian iPSCs with RNA-seq analysis. a** Workflow of the RNA-seq analysis. **b** Number of sequence reads for each sample obtained after aligning the chicken draft genome with CLC workbench. **c** Mapping ratio of each sample. Blue bars show reads mapped in pairs, red bars show reads mapped in broken pairs, and green bars show reads not mapped. **d** PCA with profiling of endangered avian and chicken iPSCs, endangered avian and chicken fibroblasts, and chicken embryo stage X. **e** Dendrogram and heat map analysis with profiling of endangered avian and chicken iPSCs, endangered avian and chicken fibroblasts, and chicken embryo stage X. **f** PCA analysis with profiling of chicken iPSCs (PB-TAD-7F), chicken fibroblasts, and chicken embryo stage X with SRP115012 (GEO: GSE102353). **g** PCA with profiling of chicken iPSCs (with PB-TAD-7F), chicken fibroblasts, and chicken embryo stage X with SRP087639 (GEO: GSE86592) and part of SRP115012 (GEO: GSE102353).

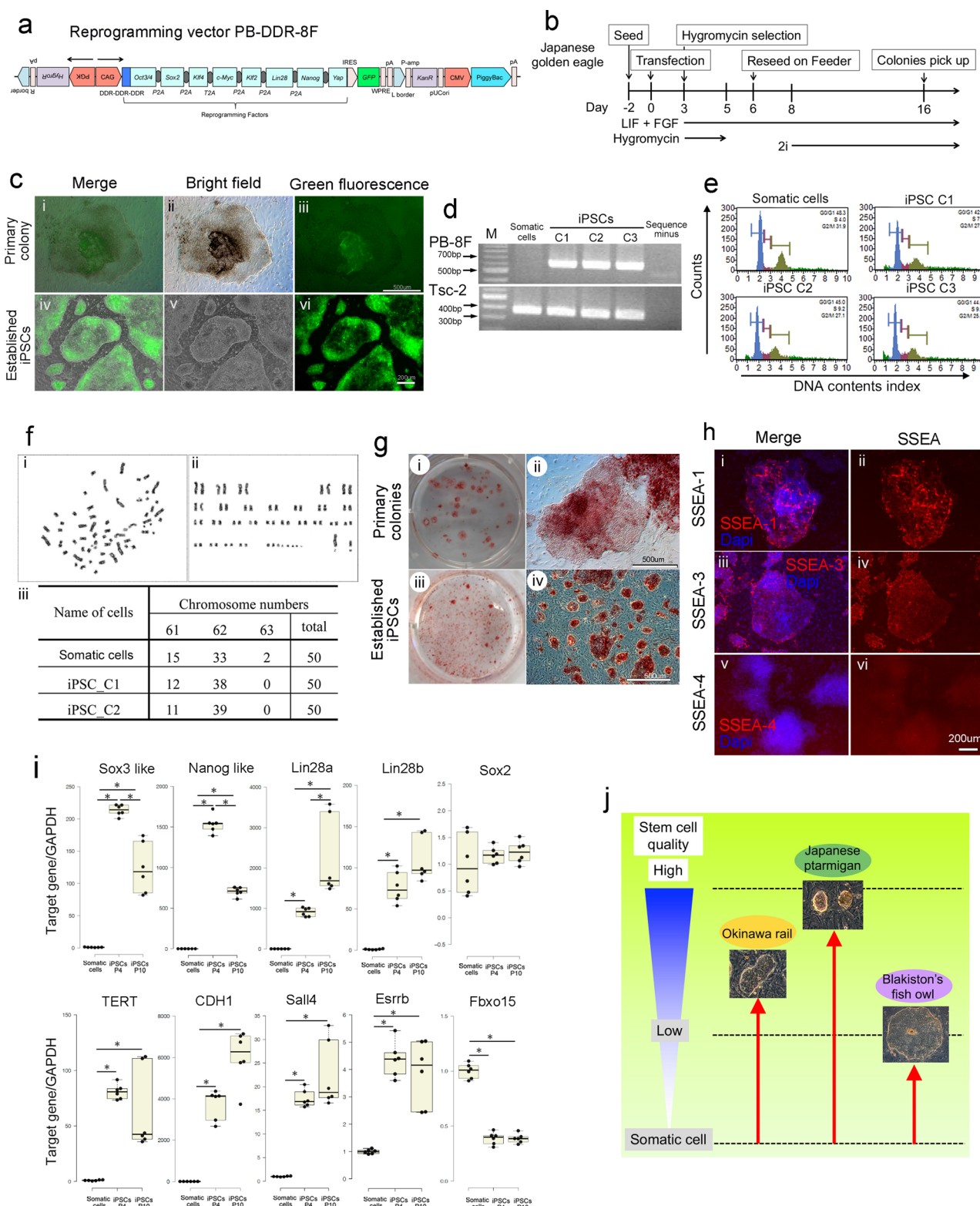

cellular characteristics. We used 1–10 μM JAK inhibitor I (4200099; MERCK, Darmstadt, Germany), 0.5–4 μM of PD173074, which inhibits FGF receptor (FGFR) inhibitor (160–26831; Wako Pure Chemical Industries), 10 μM of Y27632, which inhibits ROCK (036–24023; Wako Pure Chemical Industries), and 2 or 4 mM 2-deoxyglucose (2DG, D0051; Tokyo Chemical Industry, Tokyo, Japan).

**AP and immunological staining of fibroblasts and iPSCs.** A red-color AP staining kit (AP100 R-1; System Bioscience, Palo Alto, CA, USA) was used to detect AP activity of iPSCs. iPSCs were stained for SSEA-1, SSEA-3, and SSEA-4 antibodies (Supplementary Table 2). To stain the iPSCs with the SSEA antibodies, the

cells were fixed in 4% paraformaldehyde in phosphate buffered saline (PBS) for 3 min. Cells were permeabilized by 0.5% Triton X-100 (35501-15; Nacalai Tesque, Kyoto, Japan) for 60 min. After three washes with PBS, the iPSCs were blocked with 1% bovine serum albumin (BSA, 01863-06; Nacalai Tesque) for 45 min. iPSCs were incubated with a primary antibody overnight and then exposed to the corresponding fluorescent-labeled secondary antibodies for 60 min. Counterstaining was performed with a 4′,6-diamidino-2-phenylindole (DAPI) solution (Cellstain-DAPI solution, DOJINDO, Kumamoto, Japan).

Japanese golden eagle and chicken-derived fibroblasts were seeded in 12-well cell culture plates for immunological staining. After 48 h of incubation, F-actin

**Fig. 10 Establishment of Japanese golden eagle iPSCs from somatic cells derived from emerging pinfeather. a** Structure of the reprogramming vectors. **b** Establishment flow of Japanese golden eagle iPSCs with PB-DDR-8F. **c** Morphological features of Japanese golden eagle-derived primary and established iPSC colonies. Panels show the primary colonies of reprogrammed cells (i–iii), and established iPSC colonies (iv–vi). Panels show merged images (i, iv), bright field images (ii, v), and GFP fluorescence images (iii, vi). The bars indicate 500 μm (primary colony) and 200 μm (Established iPSCs). **d** Detection of reprogramming cassettes (PB-DDR-8F vector) and *Tsc-2* gene (internal control) using genomic PCR. **e** DNA content profile analysis of Japanese golden eagle-derived somatic cells and iPSCs (clone1–3) using flow cytometry. **f** Karyotype analysis of Japanese golden eagle-derived iPSCs. The panel shows the representative mitotic phase of the Japanese golden eagle cells (i), and the aligned chromosomes in the Japanese golden eagle cells (ii). The panel shows the chromosome numbers of the Japanese golden eagle fibroblasts and iPSCs (iii). **g** Alkaline phosphatase (AP) staining. AP staining of Japanese golden eagle-derived primary colonies (i, ii). The bar indicates 500 μm. AP staining of Japanese golden eagle-derived iPSCs (iii, iv). The bar indicates 500 μm. **h** Expression of pluripotency markers in iPSCs. Panels show SSEA-1 (i, ii), SSEA-3 (iii, iv), and SSEA-4 (v, vi). Merged images (i, iii, v), SSEA-derived fluorescence images (ii, iv, vi). The bar indicates 200 μm. **i** Pluripotency-related gene expression in Japanese golden eagle-derived somatic cells and iPSCs (passages 4 and 10). Centerlines of box plots indicate medians; box limits indicate the 25th and 75th percentiles. *n* = 6. \**P* < 0.05. **j** Study summary.

staining was performed using Alexa Fluor 568 phalloidin (A12380; Thermo Fisher Scientific) according to the manufacturer's protocol. Double staining was performed with an anti-vimentin antibody (MA5-11883; Thermo Fisher Scientific) and Alexa Fluor 488-labeled secondary antibody (A-11001; Thermo Fisher Scientific) (Supplementary Table 2). The samples were counterstained with Cellstain-DAPI solution (DOJINDO) as described above.

**Detection of reprogramming vectors and internal control genes from iPSCs.** DNA was isolated using the EZ1 DNA Tissue Kit (953034; QIAGEN, Hilden, Germany). PCR was performed with 100 ng of template DNA. Primer sequences are listed in Supplementary Tables 3 and 4. We performed PCR assays using KOD FX Neo (KFX-201; TOYOBO, Osaka, Japan). PCR was conducted by predenaturation at 94 °C for 2 min, denaturation at 98 °C for 10 s, and extension at 68 °C for 30 s, with 40 cycles of denaturation and extension. PCR products were analyzed by electrophoresis on 2.0% agarose/Tris-acetate–ethylenediaminetetraacetic acid (EDTA) gels.

**Sequential passaging.** Mouse, Okinawa rail, and Japanese ptarmigan-derived primary cells and iPSCs were seeded in six-well plates with feeder cells for analysis. When cell growth became confluent, all cells and the number of cells per dish was enumerated using a Countess cell counter (Thermo Fisher Scientific). The harvested and seeded cell numbers were used to calculate the PD time as an indicator of the speed of cell growth, using the formula PD = log2 (A/B), where A is the number of harvested cells at the end of each passage, and B is the number of seeded cells at the start[25].

**Detection of mRNA expression.** Total RNA was isolated from iPSCs using an EZ1 RNA Tissue Mini Kit (959034; QIAGEN). cDNA was synthesized from total RNA using the PrimeScript reverse transcription (RT) reagent kit (Perfect Real Time, RR047A; TaKaRa Bio, Ohtsu, Japan). Real-time PCR was performed in a 12.5 μl volume containing 2 × KOD SYBR qPCR Mix (QKD-201; Toyobo), 10 ng of cDNA solution, and 0.3 μM of each primer. The primer sequences are listed in Supplementary Tables 5–10. The reaction was performed in duplicate. The cycling program was as follows: 98 °C for 120 s (initial denaturation), 98 °C for 10 s (denaturation), 58 °C for 10 s (annealing), and 68 °C for 32 s (extension) for 40 cycles. We normalized the expression levels of the target genes to that of glyceraldehyde-3-phosphate dehydrogenase (*GAPDH*).

**Mitochondria staining.** Mitochondria were stained by incubation with 50 nM MitoTracker Orange (M7510; Thermo Fisher Scientific) or 20 nM tetramethyl rhodamine ethyl ester perchlorate (TMRE, T669; Thermo Fisher Scientific) for 10 min. After staining, the solution was removed, and fresh medium was added for observation.

**EB formation and in vitro differentiation.** In vitro differentiation of Okinawa rail, Japanese ptarmigan, Blakiston's fish owl, and Japanese golden eagle iPSCs was performed. To generate EBs, iPSCs were seeded in low-binding dishes in KAv-1 medium. After 7–14 days, floating EBs were selected and seeded in 0.1% gelatin-coated 6-well plates with KAv-1 medium. To induce differentiation into neural cells, the floating EBs were cultured in 0.1% gelatin-coated plates containing KAv-1 supplemented with 10 μM ATRA and 4.0 ng/ml FGF for 7 days.

Cells were immunochemically stained after in vitro differentiation using antibody to TUJ1, alpha-smooth muscle, or Gata4 (Supplementary Table 2). Differentiated cells were stained based on the immunological staining procedure of iPSCs described above.

**Teratoma formation and tissue sectioning.** The Animal Committee of Iwate University approved the experimental protocol for teratoma formation (approval numbers A201734, A201737). For teratoma formation, 1 × 10⁶ iPSCs were injected into the testes of SCID mice (C.B-17/Icr-scid/scidJcl; CLEA Japan, Tokyo, Japan). After 4–34 weeks post-injection, tumor tissues were excised from the mice. Each

tumor tissue was fixed with 10% formaldehyde in PBS. Fixed tissue sections were stained with hematoxylin-eosin (HE) and observed by microscopy.

Immunological staining was performed in addition to HE staining. For immunological staining, antibody to TUJ1, alpha-smooth muscle, or Gata4 was used (Supplementary Table 2). The paraffin block of each teratoma was sliced to produce a section 5 μm thick. After deparaffinization, the antigen was activated with citric acid buffer (SignalStain Citrate Unmasking Solution (10×), 14746; Cell Signaling Technology, Beverly, MA, USA) by microwaving for 10 min. To block endogenous peroxidase, tissue sections were incubated with 3% hydrogen peroxide (081–04215; Wako Pure Chemical). After washing with purified water, the tissue sections were incubated with 5% goat serum (555–76251; Wako Pure Chemical) in PBS. Next, the section were incubated in a solution containing a 1:100 dilution of primary antibody overnight at 4 °C. After washing with PBS, the tissue sections were incubated with horseradish peroxidase (HRP) conjugated secondary antibody (anti-IgG (H+L chain), mouse, pAb-HRP, code no. 330; MBL Co., Ltd., Nagoya, Japan) or anti-IgG (H+L chain, rabbit, pAb-HRP, code no. 458; MBL) for 1 h (Supplementary Table 2). After washing with PBS, the tissue sections were incubated with 3,3′-diaminobenzidine substrate solution (Histostar, code no. 8469; MBL) for 5–20 min. After washing with purified water, tissue sections were counterstained with hematoxylin for 1–2 min.

**DNA component analysis.** Cultured cells fixed with 70% ethanol at least 4 h under −20 °C condition. The fixed cells stained with the Muse Cell Cycle Assay Kit (Merck Millipore Corporation, Darmstadt, Germany). The stained cells analyzed with Muse Cell Analyzer (Merck Millipore Corporation) were used for DNA content analysis.

**Karyotype analysis.** Our iPSCs were treated with 0.02 mg/ml colcemid. Those iPSCs exposed to a hypotonic solution and fixed with Carnoy's fluid. We counted the chromosomal number in 50 cells and performed a G-banding analysis in 20 cells[22].

**Production of interspecific chimeras and their immunological staining.** To evaluate whether iPSCs derived from Japanese ptarmigan could contribute to the generation of interspecific chimeras in chick embryos, iPSCs were stained with 10 μM CellTracker Green CMFDA (5-chloromethylfluorescein diacetate, C7025; Thermo Fisher Scientific) for 30 min. Eggs of white leghorn chicken were purchased from a local farm (Goto-furanjyo, Gifu, Japan). We injected the labeled Japanese ptarmigan iPSCs into stage X chick blastoderms and cultured the embryos[26]. To confirm the contribution of chimera, fluorescence was observed after 72 h. To analyze the tissue-level contribution of chimera, embryos on day 5. The embryos were embedded in optimal cutting temperature compound (Sakura Finetek Japan, Tokyo, Japan), frozen in liquid nitrogen, and stored at −80 °C until use. Cryosections 20 μm in thickness were prepared using a cryostat, air-dried for 30 min at room temperature, and fixed with 4% paraformaldehyde for 2 min at room temperature. After washing three times with PBS, sections were incubated with PBS containing 5% FBS for 1 h. After blocking with FBS, the sections were incubated with an anti-hygromicin resistance gene antibody (anti-HPT2; Supplementary Table 2) overnight. After washing three times with PBS, the sections were incubated with secondary antibody (goat anti-mouse IgG, Alexa Fluor 568; Supplementary Table 2) and Cellstain- DAPI solution (DOJINDO) for 1 h.

**Detection of contribution of chimera from genome.** We injected Japanese ptarmigan iPSCs (without CellTracker Green CMFDA label) into a stage X chicken blastoderm. On day 5, the entire chicken embryos were collected. The genome of each embryo was isolated using NucleoSpin Tissue (U0952S; MACHEREY-NAGEL, Düren, Germany). After collecting the chimeric genome, we detected the reprogramming vector cassette using genomic PCR analysis using 50 ng of template genome. To extend the target sequence, we used the KOD FX Neo (KFX-201; TOYOBO). Primer information is provided in Supplementary Table 11. This analysis was performed according to the manufacturer's protocol. The cycling program comprised 45 cycles of 94 °C for 120 s (initial denaturation), 98 °C for 10 s

(denaturation), and 68 °C for 50 s (annealing and extension). After PCR, 2% agarose gel electrophoresis was performed. Gels were stained with GelGreen (517–53333; Biotium, Inc., Fremont, CA, USA).

Real-time PCR was also performed to detect the contribution of chimera. The fluorescence probe and primers designed to detect chimeric contributions are summarized in Supplementary Table 12. The template was a 30 ng genome. The analysis was performed using 1 × THUNDERBIRD Probe qPCR Mix (QPS-101; TOYOBO), 0.3 μM of each primer, 0.2 μM of probe, and 1 × Rox. Fifty cycle of 95 °C for 60 s (initial denaturation), 95 °C for 15 s (denaturation), and 60 °C for 60 s (annealing and extension) were used. The expression levels of the target genes were normalized to that of chicken Tsc-2.

**RNA preparation and sequencing for RNA-seq analysis.** Total RNA from iPSCs, fibroblasts, and chicken embryo stage X was collected using NucleoSpin Tissue (740952.50; MACHEREY-NAGEL). Triplicate samples of all iPSCs, fibroblasts, and chicken embryo stage X were prepared. To prepare the library, we used the TruSeq Stranded mRNA LT Sample Prep Kit (RS-122-2101; Illumina, San Diego, CA, USA). The quality of the library was evaluated using the Qubit DNA Assay (Thermo Fisher Scientific) on a TapeStation with a D1000 screen tape (Agilent Technologies, Santa Clara, CA, USA). The cDNA samples were used for the sequencing reaction on an Illumina HiSeq X sequencing machine, resulting in more than 40 M reads with 150 bp ends for each sample, except chicken fibroblast No. 3, which displayed more than 40 M reads with 75 bp ends. To analyze the RNA-seq data, we used the CLC Genomic Workbench (CLC Bio, Aarhus, Denmark). In the trim read step, low-quality sequence with the quality score of the CLC workbench, 5′ end, 3′ end, and short sequences (shorter than 15 sequences) were removed. The trimmed sequence data were mapped onto the chicken reference genome. Gene expression data were obtained in this step. PCA was performed and a heat map created with CLC Genomic Workbench using gene expression data. In this step, normalization was automatically performed using TMM methods. To compare chicken cells, RNA-seq data from SRA (SRP115012 (GEO: GSE102353) and SRP087639 (GSE86592) were used. The RNA-seq data has been submitted to the DNA DataBank of Japan under accession number DRA013522 (Submission), PRJDB13093(BioProject), SAMD00444261–SAMD00444287 (BioSample).

**Statistics and reproducibility.** Nonparametric multiple comparison analysis used the Steal–Dwass test (Figs. 2e, 3 [Okinawa rail, Japanese ptarmigan, and Blakiston's fish owl], 4d, 4f, 5b, 5d, 5f, 5h, 10i). For nonparametric independent two-group analysis, we used the Mann–Whitney U test (Fig. 3, for mouse and chicken, and 4b). Statistically significant differences are indicated by *($p < 0.05$) and **($p < 0.01$). Sample sizes and replicates are described in legends of figures.

**Reporting summary.** Further information on research design is available in the Nature Research Reporting Summary linked to this article.

## Data availability

All data of this study are included in this published article (and its Supplementary information File). The RNA-seq data are available from the DNA DataBank of Japan under accession number DRA013522 (Submission), PRJDB13093 (BioProject), SAMD00444261–SAMD00444287 (BioSample). Uncropped gels of Figs. 2a, 8f, and 10d shows in Supplementary Figs. 5, 16, and 18c. All source data underlying the graphs presented in the main figures are available as Supplementary Data. Our novel plasmids are available from the corresponding author on reasonable request after the official agreement of material transfer through the technology transfer office of Iwate university.

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

## Acknowledgements

We thank all members of the time-capsule team at the National Institute of Environmental Studies. This study was supported by the JSPS KAKENHI.

## Author contributions

M.K. was involved in the study design, collection, and/or assembly of data; data analysis and interpretation; manuscript writing (original draft and editing); and final approval of the manuscript. T.F. provided experimental material and was involved in the study design, manuscript writing (review and editing), and final approval of the manuscript. T.K. provided technical support (teratoma formation). Y.N. provided technical support (teratoma formation). A.T. provided experimental material (chick embryo) and technical support (chimera formation). M.N. provided technical support (chimera formation). H.O. provided genomic information (Okinawa rail). D.E. provided genomic information (Okinawa rail). M.A. provided experimental material (Japanese ptarmigan). Takashi Nagamine provided experimental material (Okinawa rail). Yumiko Nakaya provided experimental material (Okinawa rail). K.S. provided experimental material. Y.W.

provided experimental material. T.T. provided genome information for DDR. M.I.-M. was involved in teratoma formation. N.N. provided genomic information. M.O. provided the experimental material (endangered avian cells) and was involved in the study design and final approval of the manuscript.

## Competing interests

The authors declare no competing interests.
