## [Peer Review File · Communications Biology]

Reviewers' comments:

Reviewer #1 (Remarks to the Author):

The manuscript presents the generation of iPSCs from avian species, including the generation of putative chimeras.

It discusses the importance of creating a cell bank in order to maintain biodiversity, in special regarding endangered animals. The authors explain the importance of such an initiative in Japan, and they point out that generating germ cells is one of the advantages of having iPSCs. However attention was brought to germ cells, it was not the focus of the work. Other aspects could also be included as important.

The manuscript brings new and important data to the regenerative field, and in special, to the conservation of species. However, the paper is extremely long, and the experimental design lacks clarity. Results, methodology and discussion are unclear and most of the time repeatedly and incomplete.

How many replicates were performed? What was the efficiency in each? How many clonal colonies were generated and for how many passages? This is important once each lineage may present different characteristics, even when comparing the same species.

About the histology - immunostaining is needed to confirm possible tissue differentiation.

Points to be revised and considered:

- the abstract is not informative as presented. More results should be described even if concisely.
- On page 3 and others the paragraph is extensive. Divide the paragraphs.
- "characterization of the quality of avian iPSCs is...". This sentence must be clarified or re-written. I do not agree with iPSCs "quality", only whether they are reprogrammed or partially, and which pathways were activated or repressed.
- Line 91: "We can obtain the ..." - revise this sentence, the authors may reference the protocol and briefly describe it without using 1st person.
- "Detailed information about the sampling of emerging pinfeathers is described in our 94 other manuscript, which has been submitted to the journal at the same time"- please revise the entire text, making sure these sentences are re-written in a more formal manner. Several paragraphs contain colloquial sentences, not suitable for this journal.
- Lines 97-102: This paragraph does not present results.
- What was the lipofection efficiency? How long was the hygromycin selection period? Describe in more detail the lipofection procedure.
- Lines 600-601: "To analyze the cellular character, we used": I cannot understand. What is a cellular character, which test was used?
- Include the information in the text and cite the reference. Do not use "We described the immunological staining procedure in our previous report".
- "Genomic PCR"- seems a wrong title. Replace by an informative sentence/title. Which genes did you look for? It should be written.
- correct the spelling of germline throughout the text, it must be consistent.
- The karyotyping - why so divergent between analysis - which is normal for each, and why so many metaphases containing a different number for each species?
- PouV - correct this - POU5.
- The topic titles in the results are not adequate for each part of the text.
- In discussion: "In this study, after 446 numerous trials and errors"- which were they? This is very important information that will strengthen the study.

Reviewer #2 (Remarks to the Author):

Developing strategies to apply stem cell technology the conservation of endangered species is challenging. In this regard the current paper has attempted to apply methods used to induce pluripotent stem cells in the chicken to three endangered birds. Hence, the goals is:

1) to test if Oct3/4, Sox2, Klf4, c-Myc, Nanog, and Lin28 can be used to reprogram other birds besides the chicken, and specifically using material from endangered birds.

2) to evaluate the quality of reprogramming, based upon naïve vs. primed states of iPSCs.

Fuet et al., Stem Cell Reports

2018 Nov 13;11(5):1272-1286. doi: 10.1016/j.stemcr.2018.09.005. Epub 2018 Oct 11. also used Oct3/4, Sox2, Klf4, c-Myc, Nanog, and Lin28 to generate iPSCs from chicken fibroblasts. These iPSCs were compared to chicken embryonic stem cells. Their conclusion was that while avian cells have the ability to undergo reprogramming, something was missing in generating fully naïve iPSCs.

The current paper includes the generation of mouse iPSCs and chicken iPSCs as a comparison. And specifically in Figure 10, the mouse iPSCs are viewed as the standard for a "naïve" designation. However, is this appropriate? In mammals, "naïve" refers to a state similar to the inner cell mass of a blastocyst. Is this an appropriate comparison with birds since avian embryos do not have an inner cell mass? Would it not be better to make the comparison with chicken embryonic stem cells or the central disk of the Stage V-X chicken embryo?

The transposon vector used seems to have variable expression.

In Figure 1 GFP fluorescence differs remarkably. This suggests that the reprogramming genes are not expressed consistently and could account for the variability in outcome. A comparison of Figure 1 D and E also suggests that the expression of construct is eventually turned off or reduced except for the ptarmigan. Certainly this is a source of variation in the process.

Unfortunately, the chimera formation data is disappointing.

One would expect higher level of chimerism within the embryos than that shown. In addition only 3 out of 46 embryos were chimeric at all.

Based upon the image in Figure 10 C, it appears to me that most of the cells were not incorporated into the embryo proper, but resided in extraembryonic membranes at the junction of the area opaca and the area pellucida vasculosa. Figure 10 D of the hygromycin staining is not convincing and impossible to tell the orientation of the tissue or it's location within the embryo.

My conclusion is that while the ptarmigan iPSCs appear to be the most "naïve" compared to mouse iPSCs, they are poorly functional in situ. This really indicates that using mouse as a standard for avian iPSCs is problematic.

The Discussion addresses the fact that the extensive evaluation of the pluripotent genes among the species did not agree, meaning the authors could not find a uniform pluripotency signature for birds. From this the authors assume that the "the network of pluripotency-related genes has developed diversity in the avian evolution". A more likely case is that our knowledge of the pluripotency in birds is currently limited.

Reviewer #3 (Remarks to the Author):

In this manuscript, the authors established and identified iPSCs of three endangered avian species,

and the provided data supported it well. Therefore, the reviewer thinks that this manuscript is sufficient to be published with some minor modifications listed below.

Comments:

1. The history of avian stem cell research provided in this paper is insufficient, and for the readers' understanding, more literature should be cited and discussed in the Introduction section as follows.
 - Kim et al. Induction of Pluripotent Stem Cell-like Cells from Chicken Feather Follicle Cells. *J Anim Sci* 95(8):3479-3486 (2017).
 - Park et al. Derivation and characterization of pluripotent embryonic germ cells in chicken. *Mol Reprod Dev* 56(4):475-82 (2003).
2. References to the lines 55-57 and lines 66-74 of the Introduction section should be provided.
3. Regarding lines 93-94 of the result part, reviewer is unable to access detailed information. If you want to leave out method, the published paper should be cited.
4. For the readability of the paper, it is suggested to compose figures according to the order of results. For example, in the current version of manuscript, a supplementary figure related to second results section is organized in Figure S11.
5. In the legend of Figure S11 (Genomic PCR results), it is described as follows. Detailed information can be found in Fig. 2d (high magnification image of this gel). However, Figure 2d shows the staining results of SSEA-3 antibody. It needs to be corrected.
6. The spelling of all supplemental tables should be corrected. Fowerd (x) → Forward (o)
7. The authors used cryopreserved cells from three endangered avian species but did not describe the cryopreservation method. A detailed method for cryopreservation is required.

<Reviewer 1>

Comments for the author:

Reviewer #1 (Remarks to the Author):

The manuscript presents the generation of iPSCs from avian species, including the generation of putative chimeras.

It discusses the importance of creating a cell bank to maintain biodiversity, in special regarding endangered animals. The authors explain the importance of such an initiative in Japan, and they point out that generating germ cells is one of the advantages of having iPSCs. However attention was brought to germ cells, it was not the focus of the work. Other aspects could also be included as important.

The manuscript brings new and important data to the regenerative field, and in special, to the conservation of species. However, the paper is extremely long, and the

experimental design lacks clarity. Results, methodology and discussion are unclear and most of the time repeatedly and incomplete.

How many replicates were performed? What was the efficiency in each? How many clonal colonies were generated and for how many passages? This is important once each lineage may present different characteristics, even when comparing the same species.

About the histology - immunostaining is needed to confirm possible tissue differentiation.

<Response>

Thank you for your positive comments on our study, and we are delighted that you recognize the importance of our manuscript. According to your comments, we have replaced the text for a complete understanding of the general readers, and we have deleted the description about the possibility of contribution to germ cells. In addition to revising the manuscript concerning clarity of the language we have shown the experimental design (Fig. 2C).

Fig. S2A and main text (page 7 lines 91–97, and page 8, lines 103–109) provide information concerning the number of replicates, efficiency in each, number of clonal colonies generated and for how many passages. We established iPSCs with a PB-TAD-7F reprogramming vector, and replication was performed once for Blakiston’s fish owl, twice for Okinawa rail, three times for Japanese ptarmigan (Fig. S2A, and page 7 lines 91–97). The efficacy of the establishment of iPSCs with PB-TAD-7F is listed below as well as in Fig. S2 and in the main text on page 8, lines 103–105.

For chicken, eight primary colonies were picked and eight clones were established. For Okinawa rail, 19 primary colonies were picked and eight clones established in the first shot, and 24 primary colonies were picked, and 24 clones established in the second shot. For Japanese ptarmigans, eight primary colonies were picked and five clones were established in the first shot, and 32 primary colonies were picked, 27 clones established in the second shot, and 15 primary colonies were picked and 11 clones were established in the third shot. For Blakiston’s fish owl, seven primary colonies were picked and five clones were established.

In this study, we were able to obtain at least 20 avian iPSC primary colonies in Okinawa rail, Japanese ptarmigan, and chicken in six-well plates, while Blakiston’s fish owl had only seven primary colonies (page 7, lines 99–101), although we used an identical expression vector, indicating that there is a species difference in the efficiency of iPSC production between other avians (ex. Okinawa rail) and Blakiston’s fish owls. Our iPSCs could be passaged at least 20 times (page 8,

lines 108–109).

Furthermore, according to your advice, we detected the differentiation of iPSCs using immunostaining (Fig. 11). According to our immunostaining and histological analysis, our endangered avian iPSCs have the potential of differentiating to the three-germ layer.

Points to be revised and considered:

The abstract is not informative as presented. More results should be described even if concisely.

<Response>

We have rewritten the abstract and describe the results in more detail (page 2, lines 11–13, 15–16).

On page 3 and others the paragraph is extensive. Divide the paragraphs.

<Response>

We have divided the paragraphs and deleted the first half paragraph to shorten the text for the benefit of the readers (page 3, line 32 to page 3, line 36).

"characterization of the quality of avian iPSCs is...". This sentence must be clarified or re-written. I do not agree with iPSCs "quality", only whether they are reprogrammed or partially, and which pathways were activated or repressed.

<Response>

Thank you for your comments. According to your and Reviewer 2 's comments, we have deleted these paragraphs.

Line 91: "We can obtain the ..." - revise this sentence, the authors may reference the protocol and briefly describe it without using 1st person.

<Response>

We have corrected this sentence (page 5 line 63–64) and provide detailed information (pages 5–6, lines 64–71). Furthermore, based on the office comments, we have provided a detailed protocol (Fig.1B).

"Detailed information about the sampling of emerging pinfeathers is described in our other manuscript, which has been submitted to the journal at the same time" - please revise the entire text, making sure these sentences are re-written in a more formal manner. Several paragraphs

contain colloquial sentences, not suitable for this journal.

<Response>

We have replaced these sentences (pages 5–6, lines 63–71). Furthermore, we have again used an editing service to strength the clarity of the language.

Lines 97-102: This paragraph does not present results.

<Response>

We have deleted this paragraph from the results (pages 6) according to your comment.

What was the lipofection efficiency? How long was the hygromycin selection period? Describe in more detail the lipofection procedure.

<Response>

Lipofection efficiency was relatively low (approximately 5% or less). Therefore, we performed hygromycin selection to exclude wild-type cells. The term hygromycin selection is shown in Fig. 2C. We have shown the corresponding information (pages 7, lines 94–97, and Fig. 2C).

Lines 600-601: "To analyze the cellular character, we used": I cannot understand. What is a cellular character, which test was used?

<Response>

Thank you for your comment. We replaced this sentence to “To analyze the cellular characteristics, we focused on the Janus kinase (JAK), FGF, ROCK, and glycolytic pathways, since the dependency of these pathways can indicate differences in cellular characteristics” (page 40, line 608 to 611).

Include the information in the text and cite the reference. Do not use "We described the immunological staining procedure in our previous report".

<Response>

In accordance with your comment, we have described the immunological staining procedure in the text (page 40-41, lines 621–629).

"Genomic PCR"- seems a wrong title. Replace by an informative sentence/title. Which genes did you look for? It should be written.

<Response>

We replaced the title of “Genomic PCR” with “Detection of reprogramming vectors and

internal control genes from iPSCs”. We aimed to detect the reprogramming vector and internal control and so replaced this information in the text (page 41, line 638).

correct the spelling of germline throughout the text, it must be consistent.

<Response>

Done.

The karyotyping - why so divergent between analysis - which is normal for each, and why so many metaphases containing a different number for each species?

<Response>

We previously found that the normal karyotype of Okinawa rail would be $2n=70-78$, whereas the normal karyotype of other species is not known. In general, precise karyotype analysis of avians is more difficult than that of mammals because avians contain many microchromosomes. Therefore, our karyotype analysis data indicated that our established cells maintained diploidy (Fig.4C).

In this study, we tried to analyze whether the chromosomes of our iPSCs were diploid using DNA content analysis (Fig. 4A). The diploid clone (Fig. 4B, $2n$; black bar) histograms are shown in the middle of Fig. 4A. These histograms show the peaks of $2n$ (G1/G0 of diploids) and $4n$ (Fig. 4B, G2/metaphase of diploids) chromosomes. The tetraploid clone (Fig. 4B, $4n$; white bar) histograms are shown in the right portion of Fig. 4A. These histograms show the peaks of the $4n$ (G1/G0 of tetraploid) and $8n$ (G2/metaphase of tetraploid) chromosomes. Although raw data are not shown in this figure, we observed a triple peak ($2n$, $4n$, $8n$) in the Okinawa rail and Japanese ptarmigan histograms. We considered these clones to be a mixture of diploid and tetraploid chromosomes.

Based on these histograms, we determined whether the established iPSCs maintained diploidy. Sixteen of 27 Okinawa rail iPSCs maintained diploidy ($2n$), while 11 of 27 clones displayed tetraploid chromosomes ($2n$ and $4n$, or $4n$). In Japanese ptarmigan iPSCs, 13 of 40 clones maintained diploidy ($2n$), while all three Blakiston’s fish owl iPSC clones maintained diploidy ($2n$). The details are shown in Fig. 4B, which displays the number of maintained diploid clones ($2n$; black bar), tetraploid clones ($4n$; white bar), and a mixture of diploid and tetraploid clones ($2n$, $4n$; gray bar).

In this analysis, the difference of $4n$ is diploid or tetraploid clones.

PouV - correct this - POU5.

<Response>

Done.

The topic titles in the results are not adequate for each part of the text.

<Response>

We corrected the topic titles of the results.

In discussion: "In this study, after numerous trials and errors"- which were they? This is very important information that will strengthen the study.

<Response>

According to your advice, we have provided detailed information to explain the experimental conditions (page 28, lines 429–435).

<Reviewer 2>

Reviewer #2 (Remarks to the Author):

Developing strategies to apply stem cell technology the conservation of endangered species is challenging. In this regard the current paper has attempted to apply methods used to induce pluripotent stem cells in the chicken to three endangered birds. Hence, the goals is:

- 1) to test if Oct3/4, Sox2, Klf4, c-Myc, Nanog, and Lin28 can be used to reprogram other birds besides the chicken, and specifically using material from endangered birds.**
- 2) to evaluate the quality of reprogramming, based upon naïve vs. primed states of iPSCs.**

Fuet et al., Stem Cell Reports

2018 Nov 13;11(5):1272-1286. doi: 10.1016/j.stemcr.2018.09.005. Epub 2018 Oct 11. also used Oct3/4, Sox2, Klf4, c-Myc, Nanog, and Lin28 to generate iPSCs from chicken fibroblasts. These iPSCs were compared to chicken embryonic stem cells. Their conclusion was that while avian cells have the ability to undergo reprogramming, something was missing in generating fully naïve iPSCs.

The current paper includes the generation of mouse iPSCs and chicken iPSCs as a comparison. And specifically in Figure 10, the mouse iPSCs are viewed as the standard for a "naïve" designation. However, is this appropriate? In mammals, "naïve" refers to a state similar to the inner cell mass of a blastocyst. Is this an appropriate comparison with birds since avian embryos do not have an inner cell mass? Would it not be better to make the comparison with chicken embryonic stem cells or the central disk of the Stage V-X chicken embryo?

<Response>

Thank you for your comments. In addition to mouse and chicken iPSCs, according to your comments, we compared the endangered avian iPSCs with stage X chicken embryos (Fig. 13D, E). The collection of stage V-X requires the authorization for the animal experiments, as pregnant chicks need to be sacrificed. Although we are trying to obtain authorization for animal experiments, these processes require paperwork and inspection by the committee. In this study, whole-chicken stage X tissue was used as an alternative. Our chicken stage X profiling was close to stage VIII and stage X. Therefore, we consider that our obtained chicken stage X tissue would be an alternative material to avian iPSCs.

Although genetic differences among chicken and three endangered avians would be large for PCA analysis, we observed a tendency for the profile of Japanese ptarmigan, Okinawa rail, and Blakiston's fish owl iPSCs to move to chicken stage X from fibroblasts in the PCA (Fig. 13D).

The transposon vector used seems to have variable expression.

In Figure 1 GFP fluorescence differs remarkably. This suggests that the reprogramming genes are not expressed consistently and could account for the variability in outcome. A comparison of Figure 1 D and E also suggests that the expression of construct is eventually turned off or reduced except for the ptarmigan. Certainly this is a source of variation in the process.

<Response>

To evaluate the effect of exogenous genes on iPSCs, we compared the profiles of cESCs and our chicken iPSCs (with transactivation domain-fused Oct3/4, Sox2, Klf4, c-Myc, Nanog, Lin28, and Klf2, and containing the expression of exogenous genes) with PCA. PCA revealed the close similarity in the characteristics of our ciPSCs to the characteristics of cESCs (Fig. 13F), although our ciPSCs continuously expressed exogenous genes. Our results showed that chicken ESC data featured an almost identical cluster with iPSCs, suggesting that expression of the exogenous reprogramming vector does not affect the clustering result of whole gene expression analysis.

Unfortunately, the chimera formation data is disappointing.

One would expect higher level of chimerism within the embryos than that shown. In addition only 3 out of 46 embryos were chimeric at all.

Based upon the image in Figure 10 C, it appears to me that most of the cells were not incorporated into the embryo proper, but resided in extraembryonic membranes at the junction of the area opaca and the area pellucida vasculosa. Figure 10 D of the hygromycin

staining is not convincing and impossible to tell the orientation of the tissue or it's location within the embryo.

<Response>

In addition to the imaging and historical results, we detected the existence of a reprogramming vector in the chimeric tissues. In this round of revision, we finished up additional injection data of Japanese ptarmigan iPSCs into chicken embryos and collected the embryo on day 5. Six of 11 embryos were detected for the presence of the vector in Japanese ptarmigan iPSCs and real-time PCR with a fluorescence probe (Fig. 12F–H). Based on these results, we conclude that Japanese ptarmigan iPSCs contribute to a portion of the chicken embryo.

Furthermore, we confirmed whether an anti-hygromycin antibody stained iPSCs. As supportive evidence of our hypothesis, we confirmed that the anti-hygromycin antibody reacted with iPSCs (Fig. 12D). Therefore, we conclude that our anti-hygromycin antibody allowed us to detect chimera in chick embryos.

My conclusion is that while the ptarmigan iPSCs appear to be the most "naïve" compared to mouse iPGCs, they are poorly functional in situ. This really indicates that using mouse as a standard for avian iPSCs is problematic.

<Response>

Based on this comment, we have changed the word choice from naïve in this manuscript.

The Discussion addresses the fact that the extensive evaluation of the pluripotent genes among the species did not agree, meaning the authors could not find a uniform pluripotency signature for birds. From this the authors assume that the "the network of pluripotency-related genes has developed diversity in the avian evolution". A more likely case is that our knowledge of the pluripotency in birds is currently limited.

<Response>

We agree with your last sentence. To the best of our knowledge, the production of high-contribution chimeras has not yet been achieved, even in chick-derived iPSCs. This manuscript is the first report of the generation of endangered avian-derived iPSCs. We attempted to address the biological nature of our established cells, even though the genome information is poorly addressed in these species. Even in the establishment of potential stem cells, the establishment of these cells and their biological characteristics still has scientific significance in trying to fill out the possibility of stem cell science.

<Reviewer 3>

Reviewer #3 (Remarks to the Author):

In this manuscript, the authors established and identified iPSCs of three endangered avian species, and the provided data supported it well. Therefore, the reviewer thinks that this manuscript is sufficient to be published with some minor modifications listed below.

<Response>

Thank you for your comments. Based on your comments, we have made minor modifications to the manuscript.

Comments:

1. The history of avian stem cell research provided in this paper is insufficient, and for the readers' understanding, more literature should be cited and discussed in the Introduction section as follows.

-. Kim et al. Induction of Pluripotent Stem Cell-like Cells from Chicken Feather Follicle Cells. J Anim Sci 95(8):3479-3486 (2017).

-. Park et al. Derivation and characterization of pluripotent embryonic germ cells in chicken. Mol Reprod Dev 56(4):475-82 (2003).

<Response>

According to your comments, we have cited those articles and discussed them in the Introduction section (page 4 lines 41–42, and lines 44–46).

2. References to the lines 55-57 and lines 66-74 of the Introduction section should be provided.

<Response>

Based on other reviewer comments, we have deleted the contents of lines 55–57 and lines 66–74.

3. Regarding lines 93-94 of the result part, reviewer is unable to access detailed information. If you want to leave out method, the published paper should be cited.

<Response>

We have included the corresponding information in the manuscript (Fig. 1B, page 5, line 63 to page 6, line 71).

4. For the readability of the paper, it is suggested to compose figures according to the order of

results. For example, in the current version of manuscript, a supplementary figure related to second results section is organized in Figure S11.

<Response>

We reordered the figures.

5. In the legend of Figure S11 (Genomic PCR results), it is described as follows. Detailed information can be found in Fig. 2d (high magnification image of this gel). However, Figure 2d shows the staining results of SSEA-3 antibody. It needs to be corrected.

<Response>

We have corrected the legend of Figure S5 in the reordered figures.

6. The spelling of all supplemental tables should be corrected. Fowerd (x) → Forward (o)

<Response>

Done (Table S2 to S7).

7. The authors used cryopreserved cells from three endangered avian species but did not describe the cryopreservation method. A detailed method for cryopreservation is required.

<Response>

We describe the detailed methods for cryopreservation in the revised manuscript (page 35, lines 542–546).

Reviewers' comments:

Reviewer #1 (Remarks to the Author):

The manuscript is now greatly improved in this revised form. The conclusions suits better the results, and methodology is better described.

The study clearly brings important novelty in terms of induced pluripotency in animals rather than mammals.

Reviewer #3 (Remarks to the Author):

The revised version of the manuscript has been well addressed the reviewer's comments, and acceptance of this manuscript is proposed.

When asked by the editorial team to comment on the authors' response to the comments raised by Reviewer 2 (who was unable to rereview):

I think the authors addressed the comments from reviewer 2 adequately. One thing, reviewer #2 pointed out that the expression of GFP fluorescence differs between species when the transposon vector is used, and that in the case of a specifically established cell line, it is eliminated or reduced except for ptarmigan. This comment pointed out the problem of variation in the experimental process. However, as a response to this, the author suggested that the effect of exogenous vector expression was not a problem because chicken iPSCs had similar characteristics to ESCs even though it continued to express exogenous genes. This may not be an enough answer to the reviewer's comment. Did the author confirm the exogenous expression of GFP by introducing only the transposon vector without a transposase vector? If so, is it possible to maintain GFP expression in the established cell lines in revised figure 2E? Additional information about the passage and culture period of the established cell line in Figure 2E should be provided.

REVISION SUMMARY

We truly appreciate the invitation to revise our manuscript entitled "Induced pluripotent stem cells of endangered avian species." by Katayama *et al.* The reviewers' comments received in the previous round of review were quite productive for us. We have listed these suggestions and our corresponding responses point-by-point below. The modifications in the main text are highlighted in green.

Reviewers' comments:

Reviewer #1 (Remarks to the Author):

The manuscript is now greatly improved in this revised form. The conclusions suits better the results, and methodology is better described.

The study clearly brings important novelty in terms of induced pluripotency in animals rather than mammals.

Reviewer #3 (Remarks to the Author):

The revised version of the manuscript has been well addressed the reviewer's comments, and acceptance of this manuscript is proposed.

When asked by the editorial team to comment on the authors' response to the comments raised by Reviewer 2 (who was unable to rereview):

I think the authors addressed the comments from reviewer 2 adequately. One thing, reviewer #2 pointed out that the expression of GFP fluorescence differs between species when the transposon vector is used, and that in the case of a specifically established cell line, it is eliminated or reduced except for ptarmigan. This comment pointed out the problem of variation in the experimental process. However, as a response to this, the author suggested that the effect of exogenous vector expression was not a problem because chicken iPSCs had similar characteristics to ESCs even though it continued to express exogenous genes. This may not be an enough answer to the reviewer's comment. Did the author confirm the exogenous expression of GFP by introducing only the transposon vector without a transposase vector? If so, is it possible to maintain GFP expression in the established cell lines in revised figure 2E? Additional information about the passage and culture period of the established cell line in Figure 2E should be provided.

<Response>

Thank you for your positive comments on our study. To establish iPSCs, we used the poly

cistronic single all-in-one vector in this study, not separate vectors (Fig.2B). Therefore, there are unable to confirm the exogenous expression of GFP without a transposase. Furthermore, we described that we used all in one type piggyBac transposon in the section of materials and method for a better understanding of the readers.

According to your comment, we described the passage and culture period of the established cell line in Figure 2E (Figure legend in Figure 2E).

<Figure2E>

Mouse: This image shows the mouse iPSCs of day 6 at passage 2.

Chicken: This image shows the chicken iPSCs of day 5 at passage 2.

Okinawa rail: This image shows the Okinawa rail iPSCs of day 3 at passage 3.

Japanese ptarmigan: This image shows the Japanese ptarmigan iPSCs of day 4 at passage 9.

Blakiston's fish owl: This image shows the Blakiston's fish owl iPSCs of day 7 at passage 10.

In addition to their information, we newly performed the sequential passage of Okinawa rail and ptarmigan iPSCs. Three clones of Okinawa rail iPSCs reduced the GFP expression during five passages (passage 5 to 9), while three clones of ptarmigan iPSCs maintained the GFP expression (passage 6 to 10) (Fig.S21). We considered that variation of expression of exogenous genes depends of the host species, not clone difference.

Furthermore, we had confirmed the endogenous pluripotency-related gene expression in passage 3 and 10 with real-time PCR in previous rounds (Fig.5). Although Okinawa rail iPSCs did not maintain the exogenous gene expression, major pluripotency-related genes, such as POU5, Nanog and Lin28 maintained the higher-level expression during passage 3 and 10. These results indicated that expression of major pluripotency marker genes continues regardless of the expression levels of exogenous genes. We therefore considered that different cellular characteristics among three endangered avian are species differences, not exogenous gene expression levels (page 31 line 465 to 475).

<Fig.S21>

A: These images show the Okinawa rail iPSCs of day 7 at passage 5.

B: These images show the Okinawa rail iPSCs of day 4 at passage 9.

C: These images show the Japanese ptarmigan iPSCs of day 7 at passage 6.

D: These images show the Japanese ptarmigan iPSCs of day 4 at passage 9.

To take the image, we waked up the stored Okinawa rail (passage 4) iPSCs and Japanese ptarmigan iPSCs (passage 5). Okinawa rail passage number 5 and Japanese ptarmigan passage number 6 are immediately passage images, therefore there were needed to form iPSCs colonies for seven days.

REVIEWERS' COMMENTS:

Reviewer #3 (Remarks to the Author):

Authors well addressed the reviewer's suggestions. The manuscript can be accepted. This study is important in avian species.

REVISION SUMMARY

We truly appreciate the invitation to final revisions for our manuscript entitled "Induced pluripotent stem cells of endangered avian species." by Katayama *et al.*

REVIEWERS' COMMENTS:

Reviewer #3 (Remarks to the Author):

Authors well addressed the reviewer's suggestions. The manuscript can be accepted. This study is important in avian species.

<Response>

We thank referees for careful reading my manuscript.